# QDROP: RANDOMLY DROPPING QUANTIZATION FOR EXTREMELY LOW-BIT POST-TRAINING QUANTIZATION

**Xiuying Wei**[1,2][*]**, Ruihao Gong**[1,2][*]**, Yuhang Li**[2]**, Xianglong Liu**[1][✉]**, Fengwei Yu**[2]
[1]State Key Lab of Software Development Environment, Beihang University, [2]SenseTime Research
{weixiuying,gongruihao,liyuhang1}@sensetime.com,xlliu@buaa.edu.cn

## ABSTRACT

Recently, post-training quantization (PTQ) has driven much attention to produce efficient neural networks without long-time retraining. Despite its low cost, current PTQ works tend to fail under the extremely low-bit setting. In this study, we pioneeringly confirm that properly incorporating activation quantization into the PTQ reconstruction benefits the final accuracy. To deeply understand the inherent reason, a theoretical framework is established, indicating that the flatness of the optimized low-bit model on calibration and test data is crucial. Based on the conclusion, a simple yet effective approach dubbed as QDROP is proposed, which randomly drops the quantization of activations during PTQ. Extensive experiments on various tasks including computer vision (image classification, object detection) and natural language processing (text classification and question answering) prove its superiority. With QDROP, the limit of PTQ is pushed to the 2-bit activation for the first time and the accuracy boost can be up to 51.49%. Without bells and whistles, QDROP establishes a new state of the art for PTQ. Our code is available at https://github.com/wimh966/QDrop and has been integrated into MQBench (https://github.com/ModelTC/MQBench).

## 1 INTRODUCTION

In recent years, deep learning has been applied to all walks of life and offered substantial convenience for people's production and activities. While the performance of deep neural networks continues to increase, the memory and computation cost also scale up fastly and bring new challenges for edge devices. Model compression techniques such as network pruning (Han et al., 2015), distillation (Hinton et al., 2015), network quantization (Jacob et al., 2018) and neural architecture search (Zoph & Le, 2016) etc., are dedicated to reduce calculation and storage overhead. In this paper, we study quantization which adopts low-bit representation for weights and activations to enable fixed-point computation and less memory space.

Based on the cost of a quantization algorithm, researchers usually divide the quantization work into two categories: (1) Quantization-Aware Training (QAT) and (2) Post-Training Quantization (PTQ). QAT finetunes a pre-trained model by leveraging the whole dataset and GPU effort. On the contrary, PTQ demands much less computation to obtain a quantized model since it does not require end-to-end training. Therefore, much attention has recently been paid to PTQ (Cai et al., 2020; Wang et al., 2020; Hubara et al., 2021; Banner et al., 2019; Nahshan et al., 2019; Zhang et al., 2021a; Li et al., 2021c) due to its low cost and easy-to-use characteristics in practice.

Traditionally, PTQ pursues accuracy by performing the rounding-to-nearest operation, which focuses on minimizing the distance from the full-precision (FP) model in parameter space. In recent progress, Nagel et al. (2020); Li et al. (2021a) considered minimizing the distance in model space, i.e. the final loss objective. They use Taylor Expansion to analyze the change of loss value and derive a method to reconstruct the pre-trained model's feature by learning the rounding scheme. Such methods are efficient and effective in 4-bit quantization and can even push the limit of weight quantization to 2-bit. However, the extremely low-bit activation quantization, which faces more challenges, still fails to achieve satisfactory accuracy. We argue that one key reason is that existing

---

[*]Equal contribution, [✉] Corresponding author.

theoretical analyses only model the weight quantization as perturbation while ignoring activation's. This will lead to the same optimized model no matter which bit the activations use, which is obviously counter-intuitive and thus causes a sub-optimal solution.

In this work, the effect of activation quantization in PTQ is deeply investigated for the first time. We empirically observe that perceiving the activation quantization benefits the extremely low-bit PTQ reconstruction and surprisingly find that only partial activation quantization is more preferable. An intuitive understanding is that incorporating activation will lead to a different optimized weight. Inspired by this, we conduct theoretical studies on how activation quantization affects the weight tuning, and the conclusion is that involving activation quantization into the reconstruction helps the flatness of model on calibration data and dropping partial quantization contributes to the flatness on test data. Motivated by both empirical and theoretical findings, we propose QDROP that randomly drops quantization during the PTQ reconstruction to pursue the flatness from a general perspective. With this simple and effective approach, we set up a new state of the art for PTQ on various tasks including image classification, object detection for computer vision, and text classification and question answering for natural language processing.

To this end, this paper makes the following contributions:

1. We confirm the benefits unprecedentedly from involving activation quantization in the PTQ reconstruction and surprisingly observe that partial involvement of activation quantization performs better than the whole.
2. A theoretical framework is established to deeply analyze the influence of incorporating activation quantization into weight tuning. Using this framework, we conclude that the flatness of the optimized low-bit model on calibration data and test data is crucial for the final accuracy.
3. Based on the empirical and theoretical analyses, we propose a simple yet effective method QDROP that achieves the flatness from a general perspective. QDROP is easy to implement and can consistently boost existing methods as a plug-and-play module for various neural networks including CNNs like ResNets and Transformers like BERT.
4. Extensive experiments on a large variety of tasks and models prove that our method set up a new state of the art for PTQ. With QDROP, the 2-bit post-training quantization becomes possible for the first time.

## 2 PRELIMINARIES

**Basic Notations.** Throughout this paper, matrices (or tensors) are marked as $X$, whereas the vectors are denoted by $x$. Sometimes we use $w$ to represent the flattened version of the weight matrix $W$. Operator $\cdot$ is marked as scalar multiplication, $\odot$ is marked as element-wise multiplication for matrices or vectors. For matrix multiplication, we denote $Wx$ as matrix-vector multiplication or $WX$ as matrix-matrix multiplication.

For a feedforward neural network with activation function, we denote it as $\mathcal{G}(w, x)$ and the loss function as $L(w, x)$, where $x$ and $w$ are the network inputs and weights, respectively. Note that we assume $x$ is sampled from training dataset $\mathcal{D}_t$, thus the final loss is denoted by $\mathbb{E}_{x \sim \mathcal{D}_t}[L(w, x)]$. For the network forward function, we can write it as:

$$z_i^{(\ell+1)} = \sum_j W_{i,j}^{(\ell)} \cdot a_j^{(\ell)}, \quad f(z_i^{(\ell+1)}) = a_i^{(\ell+1)}, \tag{1}$$

where $W_{i,j}$ denotes weight connecting the $j^{th}$ activation neuron and the $i^{th}$ output. The bracket superscript $(\ell)$ is the layer index. $f(\cdot)$ indicates the activation function.

**Post-training Quantization.** Uniform quantizer maps continuous values $x \in \mathbb{R}$ into fixed-point integers. For instance, the activation quantization function can be written as $\hat{x} = \lfloor \frac{x}{s} \rceil \cdot s$, where $\lfloor \cdot \rceil$ denotes the rounding-to-nearest operator, $s$ is the step size between two subsequent quantization levels. While rounding-to-nearest operation minimizes the mean squared error between $\hat{x}$ and $x$, the minimization of parameter space certainly cannot equal to the minimization in final task loss (Li et al., 2021a), i.e., $\mathbb{E}_{x \sim \mathcal{D}_t}[L(\hat{w}, x)]$. However, in the post-training setting, we only have a tiny subset $\mathcal{D}_c \in \mathcal{D}_t$ that only contains 1k images. Thus, it is hard to minimize the final loss objective with limited data.

Recently, a series of works (Nagel et al., 2020; Li et al., 2021a) learn to either round up or down and view the new rounding mechanism as weight perturbation, i.e., $\hat{\boldsymbol{w}} = \boldsymbol{w} + \Delta \boldsymbol{w}$. Take a pre-trained network $\mathcal{G}$ as an example, they leverage Taylor Expansion to analyze the target, which reveals the quantization interactions among weights:

$$\min_{\hat{\boldsymbol{w}}} \mathbb{E}\left[L(\hat{\boldsymbol{w}}, \boldsymbol{x}) - L(\boldsymbol{w}, \boldsymbol{x})\right] \approx \min_{\hat{\boldsymbol{w}}} \mathbb{E}\left[\frac{1}{2}\Delta \boldsymbol{w}^\top \mathbf{H}^{\boldsymbol{w}} \Delta \boldsymbol{w}\right], \tag{2}$$

where $\mathbf{H}^{\boldsymbol{w}} = \mathbb{E}_{\boldsymbol{x} \sim \mathcal{D}_t} \nabla_{\boldsymbol{w}}^2 L(\boldsymbol{w}, \boldsymbol{x})$ is the expected second-order derivative. The above objective could be transformed into the change of output weighted by the output Hessian.

$$\min_{\hat{\boldsymbol{w}}} \mathbb{E}\left[\Delta \boldsymbol{w}^\top \mathbf{H}^{\boldsymbol{w}} \Delta \boldsymbol{w}\right] \approx \min_{\hat{\boldsymbol{w}}} \mathbb{E}\left[\Delta \boldsymbol{a}^\top \mathbf{H}^{\boldsymbol{a}} \Delta \boldsymbol{a}\right] \tag{3}$$

About the above minimization, they **finetune only the weight** by reconstructing each block/layer output (See Fig. 1). But they did not explore the activation quantization during output reconstruction with only modeling weight quantization as noise. The step size for activation quantization is determined after the reconstruction stage.

Intuitively, when quantizing the activations of a full-precision model to 2-bit or 3-bit, there should be different suitable weights. However, the existing works result in the same optimized weight due to the neglect of activation quantization. Therefore, we argue that when quantizing the neural network, the noise caused by activation quantization should be considered coherently with weights.

## 3 METHODOLOGY

In this section, to reveal the influence of introducing activation quantization before output reconstruction, we first conduct empirical experiments and present two observations. Then a theoretical framework is built to investigate how the activation quantization affects the optimized weights. Last, equipped with the analysis conclusions, a simple yet effective method dubbed QDROP is proposed.

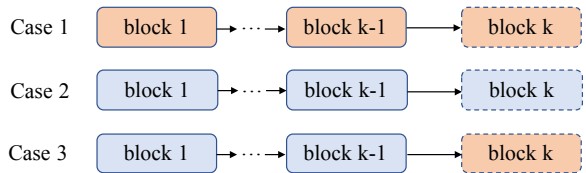

| Case | 1 | 2 | 3 |
|------|------|------|------|
| ResNet-18 | 18.88 | 45.74 | 48.07 |
| ResNet-50 | 4.34 | 46.98 | 49.07 |
| MobileNetV2 | 5.83 | 50.71 | 51.20 |
| RegNet-600MF | 42.77 | 60.94 | 62.07 |
| MnasNet | 26.62 | 58.79 | 60.19 |

Figure 1: 3 cases to involve activation quantization when optimizing the $k_{th}$ block's weight rounding. Activations are quantized inside the blue block and not quantized inside the orange block.

Table 1: 2-bit or 3-bit post-training quantization accuracy on ImageNet dataset across different cases and different models.

### 3.1 EMPIRICAL OBSERVATIONS

To investigate the influence of activation quantization when reconstructing the layer/block output, we conduct preliminary experiments on the ImageNet (Russakovsky et al., 2015) dataset. Our experiments are based on the open-sourced code Li et al. (2021a) except that we will introduce activation quantization from 1 to $k-1$ blocks before the $k_{th}$ block's reconstruction. We give a simple visualization in Fig. 1 to show 3 cases for putting activation quantization in different stages. Case 1 means that all activations are kept in 32-bit full-precision during the reconstruction of block output, which is also adopted in existing work Nagel et al. (2020); Li et al. (2021a). Case 2 and Case 3 are used for incorporating activation quantization into the reconstruction stage. However, Case 3 will omit the current block's quantization while Case 2 will not. The detailed results of these three cases are listed in Table 1 (Comparisons on 2-bit (W2A2) quantization for ResNet-18, ResNet-50 and W3A3 for others for the sake of the crashed results on 2-bit.) and the algorithm is put in algorithm 2. According to the comparison, we can obtain two observations:

1. *For extremely low-bit quantization (e.g., W2A2), there will be huge accuracy improvement when considering activation quantization during weight tuning.* This is confirmed by comparing with Case 1 and Case 2. We find Case 1 barely converges while Case 2 achieves good accuracy. It reveals that a separate optimization of weights and activations cannot find an optimal solution. After introducing the activation quantization, the weights will learn to diminish the influence of activation quantization.

2. *Partially introducing block-wise activation quantization surpasses introducing the whole activation quantization.* Case 3 does not quantize the activations inside the current tuning block but achieves better results than Case 2. This inspires us that how much activation quantization we introduce for weight tuning will affect the final accuracy.

### 3.2  HOW DOES ACTIVATION QUANTIZATION AFFECT WEIGHT TUNING

The empirical observations have highlighted the importance of activation quantization during the PTQ pipeline. To further explore how activation quantization will affect the weight tuning, we build a theoretical framework that analyzes the final loss objective with both weights and activations being quantized, which presents clues of high accuracy for extremely low-bit post-training quantization.

Conventionally, the activation quantization could be modeled as injecting some form of noise imposed on the full-precision counterpart, defined as $e = (\hat{a} - a)$. To remove the influence of activation range on $e$, we translate the noise into a multiplicative form, i.e., $\hat{a} = a \cdot (1 + u)$, where the range of $u$ is affected by bit-width and rounding error. Detailed illustration of the new form noise can be found in Appendix A.

Here, $1 + u(x)$ is adopted to present the activation noise since it is related to specific input data point $x$. Equipped with the noise, we add another argument in calculating the loss function and define our optimization objective in PTQ as:

$$\min_{\hat{w}} \mathbb{E}_{x \sim \mathcal{D}_c}[L(w + \Delta w, x, 1 + u(x)) - L(w, x, 1)]. \tag{4}$$

We hereby use a transformation that can absorb the noise on activation and transfer to weight, where the perturbation on weight is denoted as $1 + v(x)$ ($V(x)$ is used in matrix multiplication format). Consider a simple matrix-vector multiplication $Wa$ in forward pass, we have $W(a \odot (1 + u(x))) = (W \odot (1 + V(x)))a$, given by

$$W(a \odot \begin{bmatrix} 1 + u_1(x) \\ 1 + u_2(x) \\ ... \\ 1 + u_n(x) \end{bmatrix}) = (W \odot \begin{bmatrix} 1 + u_1(x) & 1 + u_2(x) & ... & 1 + u_n(x) \\ 1 + u_1(x) & 1 + u_2(x) & ... & 1 + u_n(x) \\ ... \\ 1 + u_1(x) & 1 + u_2(x) & ... & 1 + u_n(x) \end{bmatrix})a. \tag{5}$$

By taking $V_{i,j}(x) = u_j(x)$, quantization noise on the activation vector $(1 + u(x))$ can be transplanted into perturbation on weight $(1 + v(x))$. Note that for a specific input data point $x$, there are two distinct $u(x)$ and $v(x)$. Proof is available at Sec. B.1.

Also note that for a convolutional layer, we cannot apply such transformation since the input to convolution is a matrix and will cause different $V$. Nonetheless, we can give a formal lemma that absorbs $u(x)$ and holds corresponding $v(x)$ (See the Appendix Sec. B.2 for rigorous proof):

**Lemma 1.** *For a quantized (convolutional) neural network, the influence of activation quantization on the final loss objective in post-training quantization can be transformed into weight perturbation.*

$$\mathbb{E}_{x \sim \mathcal{D}_c}[L(\hat{w}, x, 1 + u(x)) - L(w, x, 1)] \approx \mathbb{E}_{x \sim \mathcal{D}_c}[L(\hat{w} \odot (1 + v(x)), x, 1) - L(w, x, 1)] \tag{6}$$

By interpolating $L(\hat{w}, x, 1)$ into Lemma 1, we can obtain the final theorem:

**Theorem 1.** *For a neural network $\mathcal{G}$ with quantized weight $\hat{w}$ and activation perturbation $1 + u(x)$, we have:*

$$\mathbb{E}_{x \sim \mathcal{D}_c}[L(\hat{w}, x, 1 + u(x)) - L(w, x, 1)] \approx$$
$$\mathbb{E}_{x \sim \mathcal{D}_c}[\underbrace{(L(\hat{w}, x, 1) - L(w, x, 1))}_{(7-1)} + \underbrace{(L(\hat{w} \odot (1 + v(x)), x, 1) - L(\hat{w}, x, 1))}_{(7-2)}]. \tag{7}$$

Here, Theorem 1 divides optimization objective into two terms. Term (7-1) is the same as Eq. (2) explored in (Nagel et al., 2020; Li et al., 2021a), which reveals how weight quantization interacts with loss function. Term (7-2) is the additional loss change by introducing activation quantization. In another way to interpret Eq. (7), the term (7-2) stands for the loss change with jitters on the weight quantized network $\mathcal{G}(\hat{w}, x)$. This type of noise correlates with certain kinds of robustness.

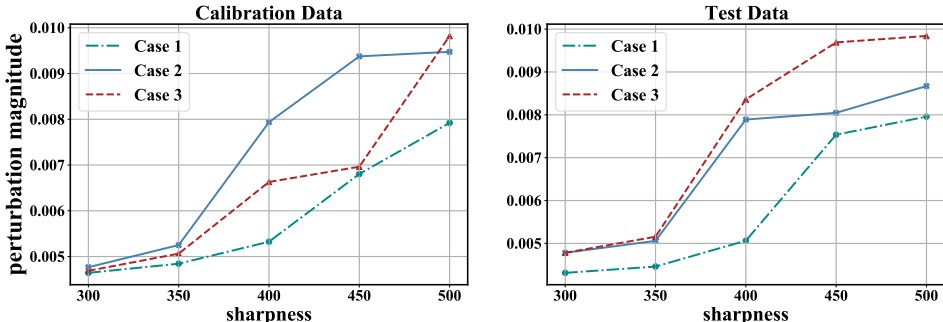

Figure 2: Measure sharpness on different data distributions among three cases. We adopt the measurement defined in (Keskar et al., 2016). With the same degree of loss change ratio, those who can tolerate a larger perturbation magnitude enjoy a flatter loss landscape.

As stated in some works about generalization and flatness (Dinh et al., 2017; Hochreiter & Schmidhuber, 1997), intuitively, flat minimum means relatively small loss change under perturbation in the parameters, otherwise, the minimum is sharp. In this paper, we follow the notion of flatness defined in (Neyshabur et al., 2017), which considers loss change from the perspective of statistical expectation. And as (Neyshabur et al., 2017) and (Jiang et al., 2019) refer to, we consider the magnitude of the perturbation with respect to the magnitude of parameters and take the formulation as $\mathbb{E}_{\boldsymbol{v} \sim \mathcal{D}}[L(f_{\boldsymbol{w} \odot (\mathbf{1}+\boldsymbol{v})}) - L(f_{\boldsymbol{w}})]$, where each element of $\boldsymbol{v}$ is a random variable sampled from a noise distribution $\mathcal{D}$ and $L$ represents for optimization objective on the training set. From this perspective, the term (7-2) can be interpreted as the flatness with perturbation related to input data, and thereby we could achieve the following corollary.

**Corollary 1.** *On calibration data $\boldsymbol{x}$, with activation quantization noise $\boldsymbol{u}(\boldsymbol{x})$, there exists the corresponding weight perturbation $\boldsymbol{v}(\boldsymbol{x})$ which satisfies that the trained quantized model is flatter under the perturbation $\boldsymbol{v}(\boldsymbol{x})$.*

With Corollary 1, Case 2 and 3 discussed in Sec. 3.1 enjoy a flatter loss landscape benefited from perceiving the activation quantization. This explains their superiority compared with Case 1. The measurement of sharpness on calibration data (left part) in Fig. 2 further validates this point. With similar perturbation magnitude, Case 2 and 3 suffer less loss degradation than Case 1.

### 3.3 QDROP

As aforementioned, introducing activation quantization is theoretically proved to produce a flatter model than existing works and the directions of flatness depend on the data distribution. Since the PTQ is especially sensitive to calibration data (Yu et al., 2021), we need to transfer the investigations in Sec. 3.2 on calibration data into the test setting for a thorough understanding. In specific, we consider Eq. (7) on test set and inspect two terms separately in the following. Based on the analyses, our method QDROP will be derived to pursue an excellent performance on test data.

**Term (7-1) on test set.** As suggested in Sec. 3.2, with both quantized activations and weights, we additionally optimize the term (7-2) representing the flatness on calibration data. This term will encourage the quantized model to learn a flat minimum. As a result, the traditional objective of AdaRound (term (7-1)) can naturally generalize better for test data (i.e., $\mathbb{E}_{\boldsymbol{x} \sim \mathcal{D}_t} (L(\hat{\boldsymbol{w}}, \boldsymbol{x}, \mathbf{1}) - L(\boldsymbol{w}, \boldsymbol{x}, \mathbf{1}))$).

**Term (7-2) on test set.** Furthermore, we should also concern about the term (7-2) on test data, i.e. $\mathbb{E}_{\boldsymbol{x} \sim \mathcal{D}_t}[L(\hat{\boldsymbol{w}} \odot (\mathbf{1} + \boldsymbol{v}(\boldsymbol{x})), \boldsymbol{x}, \mathbf{1}) - L(\hat{\boldsymbol{w}}, \boldsymbol{x}, \mathbf{1})]$. As revealed in Sec. 3.2, the term (7-2) implies the flatness where its situation on calibration data has been exploited. Here, we further investigate the flatness on test samples. Note that $\boldsymbol{v}(\boldsymbol{x})$ is converted from $\boldsymbol{u}(\boldsymbol{x})$ and this activation quantization noise varies with input data. Fig. 2 shows that there is a gap between the test data and calibration data for the flatness of the 3 cases. According to Corollary 1, these 3 cases actually introduce different $\boldsymbol{u}$ mathematically and thus will result in different flatness directions, given by

$$\text{Case 1: } \boldsymbol{u} = \mathbf{0}; \text{ Case2: } \boldsymbol{u} = \frac{\hat{\boldsymbol{a}}}{\boldsymbol{a}} - 1; \text{ Case 3: } \boldsymbol{u} = \begin{cases} \frac{\hat{\boldsymbol{a}}}{\boldsymbol{a}} - 1, & \text{block}_1 \sim \text{block}_{k-1} \\ \mathbf{0}, & \text{block}_k \end{cases}. \quad (8)$$

---

**Algorithm 1:** QDROP in one block for one batch

---

**Input:** the $k_{th}$ block from layer $i$ to layer $j$, a minibatch of quantized block input $\hat{a}^{i-1}$, FP32 block input $a^{i-1}$, FP32 block output $a^j$, quantization dropping probability $p$.
{1. Forward propagation:}
During training phase, substitute $\hat{a}^{i-1}$ with corresponding $a^{i-1}$ at neuron-level with probability $p$ and mark the replaced input as $\tilde{a}^{i-1}$ ;
**for** $l = i$ *to* $j$ **do**
    $a^l \leftarrow W^l \tilde{a}^{l-1}$;
    $\hat{a}^l \leftarrow \text{Quantize}(a^l)$;
    During training phase, randomly drop some $\hat{a}^l$ with $a^l$ as defined in Eq. (9) and get $\tilde{a}^l$ ;
{2. Backward propagation:}
Compute $\Delta a^j = \tilde{a}^j - a^j$;
Tune the weight by gradient descent ;
**return** Quantized block ;

---

For Case 1, there is no activation quantization during calibration without taking flatness into account. Case 2 suggests the activation perturbation totally and therefore enjoys a good flatness on calibration data. However, due to the mismatch on calibration data and test one, it is highly possible that Case 2 causes overfitting (See Table 8 for more details). Case 3, in fact achieves the best performance by dropping some activation quantization along with a little different weight perturbation and might not be restricted to flatness on calibration data (More evidence can be found in Table 9). This inspires us to pursue a flat minimum from a general perspective, that only optimizing the target on calibration set is suboptimal to test set.

**QDROP.** Inspired by this, we propose QDROP to further increase the flatness on as many directions as possible. In particular, we randomly disable and enable the quantization of the activation each forward pass:

$$\text{QDROP} : u = \begin{cases} 0 & \text{with probability } p \\ \frac{\hat{a}}{a} - 1 & \text{with probability } 1 - p \end{cases}. \tag{9}$$

We name it QDROP because it randomly drops the quantization of activation. Theoretically, by masking some $u(x)$ randomly, QDROP can have more diverse $v(x)$ and cover more directions of flatness thus flatter on test samples, which contributes to the final high accuracy. Fig. 3 support our analysis where QDROP has smoother loss landscape than Case 3, the winner among 3 cases on test data. Meanwhile, it is indeed a fine-grained version of Case 3 since Case 3 drops the quantization in a block-wise manner, whereas our QDROP operates in an element-wise way.

**Discussions.** QDROP can be viewed as a generalized form of the existing schemes. Case 1 and 2 respectively corresponds to the dropping probability of $p = 1$ and $p = 0$. Case 3 is equivalent to setting the block being optimized with dropping probability $p = 1$ and remains the quantization of

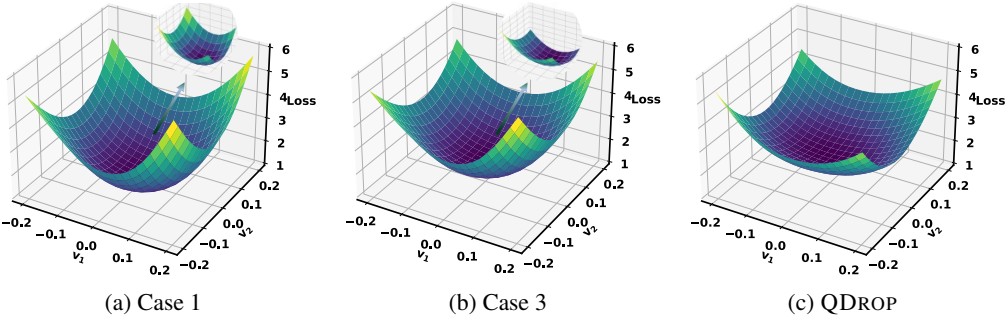

|  (a) Case 1 | (b) Case 3 | (c) QDROP |

Figure 3: Loss surface of the quantized weight for QDROP, Case 1 and 3 on test data and ResNet-18 W3A3. To better distinguish Case 1 and 3, we zoom into the local loss surface with perturbation $v_1$ and $v_2$ magnitude in [-0.025,0.025].

| Method | Bits (W/A) | ResNet-18 | ResNet-50 | MobileNetV2 | RegNet-600MF | RegNet-3.2GF | MNasNet-2.0 |
|--------|-----------|-----------|-----------|-------------|--------------|--------------|-------------|
| No Drop | 2/2 | 46.64 | 47.90 | 4.55 | 25.52 | 39.76 | 9.51 |
| QDROP | 2/2 | **51.14** | **54.74** | **8.46** | **38.90** | **52.36** | **22.70** |
| No Drop | 2/4 | 64.16 | 69.60 | 51.61 | 61.52 | 70.29 | 60.00 |
| QDROP | 2/4 | **64.66** | **70.08** | **52.92** | **63.10** | **70.95** | **62.36** |

Table 2: Effect of QDROP.

other parts. Note that the $p$ obeys Bernoulli distribution and thus can be set as 0.5 for the maximal entropy (Qin et al., 2020), which is helpful for flatness across various directions.

QDROP is easy to implement for various neural networks including CNNs and Transformers, and plug-and-play with little additional computational complexity. With QDROP, the complicated problem of choosing optimization order, i.e. different cases in Sec. 3.1, can be avoided.

## 4 EXPERIMENTS

In this section, we conduct two sets of experiments to verify the effectiveness of QDROP. In Sec. 4.1, we first conduct an ablation study for the impact with and without dropping quantization and analyze the option of distinct dropping rates. In Sec. 4.2, we compare our method with other existing approaches across vision and language tasks including image classification on ImageNet, object detection on MS COCO, and natural language processing on GLUE benchmark and SQuAD.

**Implementation Details.** Our code is based on PyTorch Paszke et al. (2019). We set the default dropping probability $p$ as 0.5, except we explicitly mention it. The weight tuning method is the same with Nagel et al. (2020); Li et al. (2021a). Each block or layer output is reconstructed for 20k iterations. For ImageNet dataset, we sample 1024 images as calibration set, while COCO we use 256 images. In NLP, we sample 1024 examples. We also keep the first and the last layer in 8-bit except NLP tasks and adopt per-channel weight quantization. We use W4A4 to represent 4-bit weight and activation quantization. More model choices and other setting is described in Appendix E.

But to be noted, regular **first and last layer 8-bit** means 8-bit weight and input in first and last layer while BRECQ uses another setting which not only keeps the first layer's input 8-bit but also the first layer's output (second layer's input). This would indeed perform better than the regular one with leaving one more layer's input 8-bit but may not be practical on the hardware. Therefore, we both compare with BRECQ's setting to show the superiority of our approach and experiment on the usual one to provide a practical baseline. Symbol † is used to mark BRECQ's setting.

### 4.1 ABLATION STUDY

**Effect of QDROP.** We propose QDROP and here we would like to test the effect of PTQ with or without QDROP. We use ImageNet classification benchmark and quantize the weight parameters to 2-bit and quantize activation to 2/4-bit. As shown in Table 2, QDROP improves the accuracy across all bit settings evaluated for 6 models on ImageNet. Furthermore, the gains are more obvious when applying QDROP to lightweight network architecture: 2.36% increment for MNasNet under W2A4 and 12.6% for RegNet-3.2GF with W2A2.

**Effect of Dropping Probability.** We also explore the dropping probability in PTQ. We choose $p$ in [0,0.25,0.5,0.75,1] and test on MobileNetV2 and RegNet-600MF. The results are summarized in Fig. 5. We find 0.5 generally performs best among 5 candidates. Although there could be a fine-grained best solution for each architecture, we shall avoid cumbersome hyper-parameter search and continue using 0.5.

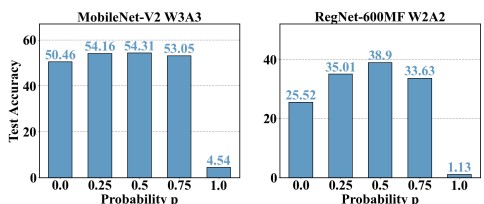

Figure 5: Impact of dropping probability on ImageNet.

### 4.2 LITERATURE COMPARISON

**ImageNet.** We choose ResNet-18 and -50 (He et al., 2016), MobileNetV2 (Sandler et al., 2018), searched MNasNet (Tan et al., 2019) and RegNet (Radosavovic et al., 2020). We summarize the

| Methods | Bits (W/A) | Res18 | Res50 | MNV2 | Reg600M | Reg3.2G | MNasx2 |
|---|---|---|---|---|---|---|---|
| Full Prec. | 32/32 | 71.06 | 77.00 | 72.49 | 73.71 | 78.36 | 76.68 |
| ACIQ-Mix (Banner et al., 2019) | 4/4 | 67.00 | 73.80 | - | - | - | - |
| ZeroQ (Cai et al., 2020)* | 4/4 | 21.71 | 2.94 | 26.24 | 28.54 | 12.24 | 3.89 |
| LAPQ (Nahshan et al., 2019) | 4/4 | 60.30 | 70.00 | 49.70 | 57.71* | 55.89* | 65.32* |
| AdaQuant (Hubara et al., 2021) | 4/4 | **69.60** | **75.90** | 47.16* | - | - | - |
| Bit-Split (Wang et al., 2020) | 4/4 | 67.56 | 73.71 | - | - | - | - |
| AdaRound (Nagel et al., 2020)* | 4/4 | 67.96 | 73.88 | 61.52 | 68.20 | 73.85 | 68.86 |
| QDROP (Ours) | 4/4 | 69.10 | 75.03 | **67.89** | **70.62** | **76.33** | **72.39** |
| AdaRound† (Nagel et al., 2020)* | 4/4 | 69.36 | 74.76 | 64.33 | - | - | - |
| BRECQ† (Li et al., 2021a) | 4/4 | 69.60 | 75.05 | 66.57 | 68.33 | 74.21 | 73.56 |
| QDROP† (Ours) | 4/4 | **69.62** | **75.45** | **68.84** | **71.18** | **76.66** | **73.71** |
| LAPQ (Nahshan et al., 2019)* | 2/4 | 0.18 | 0.14 | 0.13 | 0.17 | 0.12 | 0.18 |
| AdaQuant (Hubara et al., 2021)* | 2/4 | 0.11 | 0.12 | 0.15 | - | - | - |
| AdaRound (Nagel et al., 2020)* | 2/4 | 62.12 | 66.11 | 36.31 | 57.00 | 63.89 | 46.73 |
| QDROP (Ours) | 2/4 | **64.66** | **70.08** | **52.92** | **63.10** | **70.95** | **62.36** |
| AdaRound† (Nagel et al., 2020)* | 2/4 | 64.14 | 68.40 | 41.52 | 59.27 | 65.33 | 53.77 |
| BRECQ† (Li et al., 2021a) | 2/4 | 64.80 | 70.29 | 53.34 | 59.31 | 67.15 | 63.01 |
| QDROP† (Ours) | 2/4 | **65.25** | **70.65** | **54.22** | **63.80** | **71.70** | **64.24** |
| AdaQuant (Hubara et al., 2021)* | 3/3 | 60.09 | 67.46 | 2.23 | - | - | - |
| QDROP (Ours) | 3/3 | **65.56** | **71.07** | **54.27** | **64.53** | **71.43** | **63.47** |
| AdaRound† (Nagel et al., 2020)* | 3/3 | 64.66 | 66.66 | 15.20 | 51.01 | 56.79 | 47.89 |
| BRECQ† (Li et al., 2021a)* | 3/3 | 65.87 | 68.96 | 23.41 | 55.16 | 57.12 | 49.78 |
| QDROP† (Ours) | 3/3 | **66.75** | **72.38** | **57.98** | **65.54** | **72.51** | **66.81** |
| QDROP (Ours) | 2/2 | **51.14** | **54.74** | **8.46** | **38.90** | **52.36** | **22.70** |
| BRECQ† (Li et al., 2021a)* | 2/2 | 42.54 | 29.01 | 0.24 | 3.58 | 3.62 | 0.61 |
| QDROP† (Ours) | 2/2 | **54.72** | **58.67** | **13.05** | **41.47** | **55.11** | **28.77** |

Table 3: Comparison among different post-training quantization strategies with low-bit activation in terms of accuracy on ImageNet. * represents for our implementation according to open-source codes and † means using BRECQ's first and last layer 8-bit setting, which also keeps first layer's output 8-bit besides input and weight in the first and last layer.

results in Table 3. First, the W4A4 quantization is investigated. It can be observed that QDROP provides $0 \sim 3\%$ accuracy uplift when compared to strong baselines including AdaRound, BRECQ. As for the gap between our method and AdaQuant on W4A4, we argue that there are some discrepancies on settings such as positions of quantization nodes and put this explaination in Sec. C.3. With W2A4 quantization, QDROP can improve the accuracy of ResNet-50 by 0.5%, and RegNet-3.2GF by 4.6%. In addition, to fully exploit the limit of QDROP, we conduct more challenging cases with 2/3-bit weights and activations. According to the last two rows of Table 3, our proposed QDROP consistently achieves good results while existing methods suffer from non-negligible accuracy drop. For W3A3, the difference is even larger on MobileNetV2, where our method reaches 58% accuracy and BRECQ only gets 23%. In the W2A2 setting, the PTQ becomes much harder. QDROP outperforms the competing method by a large margin: 12.18% upswings for ResNet-18, 29.66% for ResNet-50 and 51.49% for RegNet-3.2GF.

**MS COCO.** In this part, we validate the performance of QDROP on object detection task using MS COCO dataset. We use both two-stage Faster RCNN (Ren et al., 2015) and one-stage RetinaNet (Lin et al., 2017) models. Backbone are selected from ResNet-18, ResNet-50, and MobileNetV2. Note that we set the first layer and the last layer to 8-bit and do not quantize the head of the model, however, the neck (FPN) is quantized. Experiments show that W4A4 quantization using QDROP nearly do not affect Faster RCNN's mAP. For RethinaNet, our method has 5 mAP improvement on MobileNetV2 backbone. In low bit setting W2A4, our method shows great improvement both on Faster-RCNN and RetinaNet, up to 6.5 mAP.

**GLUE benchmark and SQuAD.** We test QDROP in NLP tasks including the GLUE benchmark and SQuAD1.1. They are all conducted on the typical NLP model, i.e, BERT (Devlin et al., 2018). Compared with those QAT methods (Bai et al., 2020), which usually adopt data augmentation trick to achieve dozens of times the original data, we only randomly extract 1024 examples without any extra data processing. Besides AdaQuant and BRECQ, which suffer a huge accuracy degradation, our QDROP surpasses No Drop all the tasks, specifically on QNLI(8.7%), QQP(4.6%) and RTE(7.2%).

| Method | Bits (W/A) | Faster RCNN | | | RetinaNet | | |
|--------|-----------|-------------|--|--|-----------|--|--|
| | | ResNet-18 | ResNet-50 | MobileNetV2 | ResNet-18 | ResNet-50 | MobileNetV2 |
| Full Prec. | 32/32 | 34.60 | 38.56 | 33.47 | 33.22 | 36.80 | 32.63 |
| AdaRound* | 4/4 | 32.57 | 34.47 | 26.11 | 31.04 | 33.51 | 24.99 |
| BRECQ†* | 4/4 | 32.58 | 34.59 | 26.58 | 31.21 | 33.47 | 24.84 |
| QDROP | 4/4 | **33.37** | **36.96** | **30.88** | **31.99** | **35.67** | **29.75** |
| AdaRound* | 2/8 | 30.54 | 33.15 | 25.35 | 29.30 | 32.22 | 24.22 |
| BRECQ† | 2/8 | 31.82 | 34.23 | 27.54 | **31.42** | 34.75 | **27.59** |
| QDROP | 2/8 | **32.20** | **36.14** | **28.48** | 31.03 | **34.84** | 27.42 |
| BRECQ†* | 2/4 | 29.92 | 30.23 | 19.35 | 28.73 | 29.47 | 18.46 |
| QDROP | 2/4 | **31.01** | **34.23** | **25.04** | **29.69** | **33.01** | **24.89** |

Table 4: Comparison among typical post-training quantization strategies in terms of mAP on MS COCO. Note that refer to BRECQ, we didn't quantize head and keep the first and last layer in backbone to 8-bit. Other notations align with Table 3.

| Method | SST-2 (acc) | QNLI (acc) | QQP (f1/acc) | STS-B (Pearson/Spearman corr) | MNLI (acc m/mm) | MRPC (acc) | RTE (acc) | CoLA (Matthews corr) | SQuAD1.1 (f1) |
|--------|-------------|------------|--------------|-------------------------------|------------------|------------|-----------|----------------------|---------------|
| Full Prec. | 92.43 | 91.54 | 87.81/90.91 | 88.04/ 87.63 | 84.57/84.46 | 87.71 | 72.56 | 53.39 | 88.42 |
| AdaQuant* | - | - | - | - | - | - | - | - | 5.17 |
| BRECQ* | 50.86 | 50.72 | 4.47/62.28 | 5.94/ 6.39 | 31.91/31.81 | 31.69 | 52.34 | 0.946 | 68.58 |
| NO DROP | 87.94 | 68.05 | 68.09/76.69 | 82.24/81.68 | 69.19/71.28 | 77.39 | 53.43 | 40.17 | 75.97 |
| QDROP | **88.06** | **76.75** | **72.66/79.04** | **82.39/81.88** | **71.43/73.70** | **79.15** | **60.65** | **40.85** | **77.26** |

Table 5: Performance on NLP tasks compared to other methods on E8W4A4. Here, we use symbol EeWwAa to additionally express the embedding bit and conduct experiments on GLUE and SQuAD1.1.

As for SST-2, despite little enhancement by dropping quantization, it is indeed close to the FP32 value within 4.4%. And for STS-B, we argue that the original fine-tuned model is trained with limited data, which might not be very representative.

### 4.3 ROBUSTNESS OF QDROP

In this part, we discuss the effectiveness of QDROP under more challenging situations including even less data and cross-domain ones. Concerning about size of calibration data, we consider another 4 options. It can be observed that dropping some quantization behaves better under each setting and is even comparable with No Drop with half of the original calibration data. Motivated by Yu et al. (2021), we also reconstruct block output by 1024 examples from out-of-domain data, i.e, CIFAR100 (Krizhevsky et al., 2009), MS COCO, and test on ImageNet. Results are available in Table 6, where our QDROP still works steadily.

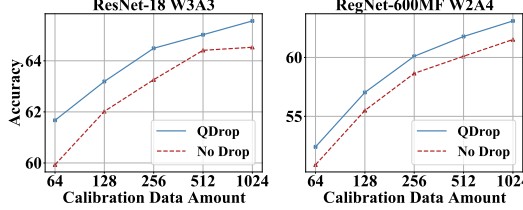

| Calibration | Bits (W/A) | No Drop | QDROP |
|-------------|-----------|---------|-------|
| MS COCO | 4/4 | 68.64 | **68.94** |
| MS COCO | 3/3 | 64.12 | **65.15** |
| CIFAR100 | 4/4 | 46.83 | **52.88** |
| CIFAR100 | 3/3 | 21.74 | **28.58** |

Figure 6: Impact of calibration data size on ImageNet.  Table 6: Cross domain data.

### 5 CONCLUSION

In this paper, we have introduced QDROP, a novel mechanism for post-training quantization. QDrop aims to achieve good test accuracy given a tiny calibration set. This is done by optimization towards a flat minima. We dissect the PTQ objective theoretically into a flatness problem and improve the flatness from a general perspective. We comprehensively verify the effectiveness of QDROP on a large variety of tasks. It can achieve a nearly lossless 4-bit quantized network and can significantly improve the 2-bit quantization results.

ACKNOWLEDGMENT

We sincerely thank the anonymous reviewers for their serious reviews and valuable suggestions to make this better. And we thank Xiangguo Zhang and Sheng Chen for their kind of help of this work. This work was supported in part by the National Natural Science Foundation of China under Grant 62022009 and Grant 61872021, the SenseTime Research Fund for Young Scholars, and the Beijing Nova Program of Science and Technology under Grant Z191100001119050.

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

## A  NOISE FORM CHOICE

As mentioned in Sec. 2, we refer the quantizer to $\hat{a} = \lfloor \frac{a}{s} \rceil \cdot s$, where $s$ is the step size that can be learned or be determined by collecting activation distribution in PTQ. By marking the rounding error as $c$, which subjects to $\mathcal{U}[-0.5, 0.5]$, the additive noise $e = \hat{a} - a$ can be denoted as $c \cdot s$. However, this traditional noise does not decouple the noise from step size $s$ and the range of activations. The range of noise $e$ will vary when the range of activations change, making it complicated to analyze across the whole network in a unified form. Some existing papers (Jiang et al., 2019) also indicate the problem that the additive noise doesn't take parameters' magnitude into account and suggest a multiplicative form form (Keskar et al., 2016). To be specific, they claim that:

> Perturbing the parameters without taking their magnitude into account can cause many of them to switch signs. Therefore, one cannot apply large perturbations to the model without changing the loss significantly. One possible modification to improve the perturbations is to choose the perturbation magnitude based on the magnitude of the parameter. In that case, it is guaranteed that if the magnitude of perturbation is less than the magnitude of the parameter, then the sign of the parameter does not change.

To eliminate the influence induced by the range of activations, we take the noise of multiplicative form: $\hat{a} = (1 + u) \cdot a$. In the quantization background, we denote the fixed-point integer value with respect to $a$ as $\bar{a}$ ($a = (\bar{a} + c) \cdot s$ and $\hat{a} = \bar{a} \cdot s$). And then $u$ can be denoted as:

$$
\begin{aligned}
u &= \frac{\hat{a}}{a} - 1 \\
&= \frac{\bar{a} \cdot s}{(\bar{a} + c) \cdot s} - 1 \\
&= \frac{\bar{a}}{\bar{a} + c} - 1 \\
&= \frac{-c}{\bar{a} + c}.
\end{aligned}
\tag{10}
$$

To be noted, $u$ is derived from the definition of the quantizer and can be equivalently transformed from the generalized noise here.

From this formulation, we can find that the range of $u$ is not related to the activation range or step size $s$ and is only influenced by the rounding error and the bit-width. In a word, the multiplicative form has its pysical meaning in quantization background and is beneficial as discussed above.

## B  PROOF OF LEMMA 1 AND THEOREM 1

We demonstrate them by considering the situation of (1) fully connected and (2) convolutional networks separately. As transformation with fully connected ones has been revealed in main body, we add some extra illustration here and mainly target at the convolutional layers.

### B.1  FULLY CONNECTED NETWORKS

Here, we give the proof of Eq. (5) by leveraging the definition of FC layers in Eq. (1). We first look at each input sample and temporarily omit $\boldsymbol{x}$ in the notation below for simplicity.

With activation noise $\boldsymbol{u}$,

$$
\begin{aligned}
\boldsymbol{z}_i^{(\ell+1)} &= \sum_j \boldsymbol{W}_{i,j}^{(\ell)} \cdot (1 + \boldsymbol{u}_j^{(\ell)}) \cdot \boldsymbol{a}_j^{(\ell)} \\
&= \sum_j (1 + \boldsymbol{u}_j^{(\ell)}) \cdot \boldsymbol{W}_{i,j}^{(\ell)} \cdot \boldsymbol{a}_j^{(\ell)}.
\end{aligned}
\tag{11}
$$

By taking $\boldsymbol{V}_{i,j}^{(\ell)} = \boldsymbol{u}_j^{(\ell)}$, we have

$$
\boldsymbol{z}_i^{(\ell+1)} = \sum_j (1 + \boldsymbol{V}_{i,j}^{(\ell)}) \cdot \boldsymbol{W}_{i,j}^{(\ell)} \cdot \boldsymbol{a}_j^{(\ell)}.
\tag{12}
$$

Thus, with proper constructed $\boldsymbol{v}$, the noise on activation ($\boldsymbol{u}$) can be viewed as the perturbation on weight ($\boldsymbol{v}$).

For every layer, we can conduct this transformation from $\boldsymbol{u}$ to $\boldsymbol{v}$. Therefore, considering operating on the quantized weight and calibration data, optimizing $\mathbb{E}_{\boldsymbol{x}\sim\mathcal{D}_c}[L(\hat{\boldsymbol{w}}, \boldsymbol{x}, \mathbf{1} + \boldsymbol{u}(\boldsymbol{x}))]$ can be approximated to optimizing $\mathbb{E}_{\boldsymbol{x}\sim\mathcal{D}_c}[L(\hat{\boldsymbol{w}} \odot (\mathbf{1} + \boldsymbol{v}(\boldsymbol{x})), \boldsymbol{x}, \mathbf{1})]$. Then we have

$$\mathbb{E}_{\boldsymbol{x}\sim\mathcal{D}_c}[L(\hat{\boldsymbol{w}}, \boldsymbol{x}, \mathbf{1} + \boldsymbol{u}(\boldsymbol{x})) - L(\boldsymbol{w}, \boldsymbol{x}, \mathbf{1})] \approx \mathbb{E}_{\boldsymbol{x}\sim\mathcal{D}_c}[L(\hat{\boldsymbol{w}} \odot (\mathbf{1} + \boldsymbol{v}(\boldsymbol{x})), \boldsymbol{x}, \mathbf{1}) - L(\boldsymbol{w}, \boldsymbol{x}, \mathbf{1})]. \quad (13)$$

Now, Lemma 1 is proved for the case of fully connected networks.

## B.2 Convolutional Networks

We first look at each input sample and temporarily omit $\boldsymbol{x}$ in the notation below for simplicity. One layer in the network $\mathcal{G}$ can be interpreted as:

$$\boldsymbol{A}_{i,j}^{(\ell+1)} = f(\boldsymbol{Z}_{i,j}^{(\ell+1)}) = f(\sum_{p,q} \boldsymbol{W}_{p,q}^{(\ell)} \cdot \boldsymbol{A}_{i+p,j+q}^{(\ell)}), \quad (14)$$

where the $f(\cdot)$ is the activation function, $(p, q)$ pair represents for one element in weight matrix and $(i, j)$ pair represents for one element in activation matrix.

Due to the complicated design of convolutional structure, based on $\mathcal{G}$ we introduce two networks $\mathcal{G}_1, \mathcal{G}_2$ to make proofs more clearly.

**Definition 1.** *$\mathcal{G}_1$ means inserting random noise on activations, $\mathcal{G}_2$ means sticking random variables into weights.*

$$\begin{aligned} \mathcal{G}_1 : \boldsymbol{A}_{i,j}^{(\ell+1)} &= f(\boldsymbol{Z}_{i,j}^{(\ell+1)}) = f(\sum_{p,q} \boldsymbol{W}_{p,q}^{(\ell)} \cdot (1 + \boldsymbol{U}_{i+p,j+q}^{(\ell)}) \cdot \boldsymbol{A}_{i+p,j+q}^{(\ell)}) \\ \mathcal{G}_2 : \boldsymbol{A}_{i,j}^{(\ell+1)} &= f(\boldsymbol{Z}_{i,j}^{(\ell+1)}) = f(\sum_{p,q} (1 + \boldsymbol{V}_{p,q}^{(\ell)}) \cdot \boldsymbol{W}_{p,q}^{(\ell)} \cdot \boldsymbol{A}_{i+p,j+q}^{(\ell)})) \end{aligned} \quad (15)$$

**Definition 2.** *We mark losses of network $\mathcal{G}_1$ and $\mathcal{G}_2$ in the following way:*

$$\begin{aligned} &\text{Loss of } \mathcal{G}_1 : L(\boldsymbol{w}, \boldsymbol{x}, \mathbf{1} + \boldsymbol{u}) &&\text{when taking } \boldsymbol{u} = \mathbf{0}, \text{it becomes } L(\boldsymbol{w}, \boldsymbol{x}, \mathbf{1}) \\ &\text{Loss of } \mathcal{G}_2 : L(\boldsymbol{w} \odot (\mathbf{1} + \boldsymbol{v}), \boldsymbol{x}) &&\text{when taking } \boldsymbol{v} = \mathbf{0}, \text{it becomes } L(\boldsymbol{w} \odot \mathbf{1}, \boldsymbol{x}). \end{aligned} \quad (16)$$

With the definitions, we first prove that $\mathcal{G}_1$ and $\mathcal{G}_2$ share common parts in their first-order derivative to its noise in Lemma 2.

**Lemma 2.** *Assuming the same weight and taking $\boldsymbol{u} = \mathbf{0}$ and $\boldsymbol{v} = \mathbf{0}$ for $\mathcal{G}_1$ and $\mathcal{G}_2$, we have*

$$\begin{aligned} \frac{\partial L(\boldsymbol{w}, \boldsymbol{x}, \mathbf{1})}{\partial \boldsymbol{U}_{i,j}^{(\ell)}} &= \sum_{p,q} \boldsymbol{T}_{(i,j),(p,q)}^{(\ell)} \\ \frac{\partial L(\boldsymbol{w} \odot \mathbf{1}, \boldsymbol{x})}{\partial \boldsymbol{V}_{p,q}^{(\ell)}} &= \sum_{i,j} \boldsymbol{T}_{(i,j),(p,q)}^{(\ell)}, \end{aligned} \quad (17)$$

*where*

$$\begin{aligned} \boldsymbol{T}_{(i,j),(p,q)}^{(\ell)} &= \frac{\partial L(\boldsymbol{w}, \boldsymbol{x}, \mathbf{1})}{\partial \boldsymbol{A}_{i-p,j-q}^{(\ell+1)}} \cdot f'(\boldsymbol{Z}_{i-p,j-q}^{(\ell+1)}) \cdot \boldsymbol{W}_{p,q}^{(\ell)} \cdot \boldsymbol{A}_{i,j}^{(\ell)} \\ &= \frac{\partial L(\boldsymbol{w} \odot \mathbf{1}, \boldsymbol{x})}{\partial \boldsymbol{A}_{i-p,j-q}^{(\ell+1)}} \cdot f'(\boldsymbol{Z}_{i-p,j-q}^{(\ell+1)}) \cdot \boldsymbol{W}_{p,q}^{(\ell)} \cdot \boldsymbol{A}_{i,j}^{(\ell)}. \end{aligned} \quad (18)$$

*Note that with $\boldsymbol{u} = \mathbf{0}$ and $\boldsymbol{v} = \mathbf{0}$, the activations are the same for the two networks so we don't use different notations here.*

*Proof.*

$$\frac{\partial L(\boldsymbol{w}, \boldsymbol{x}, \mathbf{1})}{\partial \boldsymbol{U}_{i,j}^{(\ell)}} = \sum_{p,q} \frac{\partial L(\boldsymbol{w}, \boldsymbol{x}, \mathbf{1})}{\partial \boldsymbol{A}_{i-p,j-q}^{(\ell+1)}} \cdot f'(\boldsymbol{Z}_{i-p,j-q}^{(\ell+1)}) \cdot \boldsymbol{W}_{p,q}^{(\ell)} \cdot \boldsymbol{A}_{i,j}^{(\ell)} \quad (19)$$

$$\frac{\partial L(\boldsymbol{w} \odot \mathbf{1}, \boldsymbol{x})}{\partial \boldsymbol{V}_{p,q}^{(\ell)}} = \sum_{i,j} \frac{\partial L(\boldsymbol{w} \odot \mathbf{1}, \boldsymbol{x})}{\partial \boldsymbol{A}_{i,j}^{(\ell+1)}} \cdot f'(\boldsymbol{Z}_{i,j}^{(\ell+1)}) \cdot \boldsymbol{W}_{p,q}^{(\ell)} \cdot \boldsymbol{A}_{i+p,j+q}^{(\ell)}$$
$$= \sum_{i,j} \frac{\partial L(\boldsymbol{w} \odot \mathbf{1}, \boldsymbol{x})}{\partial \boldsymbol{A}_{i-p,j-q}^{(\ell+1)}} \cdot f'(\boldsymbol{Z}_{i-p,j-q}^{(\ell+1)}) \cdot \boldsymbol{W}_{p,q}^{(\ell)} \cdot \boldsymbol{A}_{i,j}^{(\ell)} \tag{20}$$

Because of taking the same weight, and calculating the derivatives to $\boldsymbol{U}_{i,j}^{(\ell)}$ at $\boldsymbol{u} = 0$ and derivatives to $\boldsymbol{V}_{i,j}^{(\ell)}$ at $\boldsymbol{v} = 0$, we have the same activation values for these two networks. Therefore, Eq. (21) holds.

$$\frac{\partial L(\boldsymbol{w}, \boldsymbol{x}, \mathbf{1})}{\partial \boldsymbol{A}_{i-p,j-q}^{(\ell+1)}} \cdot f'(\boldsymbol{Z}_{i-p,j-q}^{(\ell+1)}) \cdot \boldsymbol{W}_{p,q}^{(\ell)} \cdot \boldsymbol{A}_{i,j}^{(\ell)} = \frac{\partial L(\boldsymbol{w} \odot \mathbf{1}, \boldsymbol{x})}{\partial \boldsymbol{A}_{i-p,j-q}^{(\ell+1)}} \cdot f'(\boldsymbol{Z}_{i-p,j-q}^{(\ell+1)}) \cdot \boldsymbol{W}_{p,q}^{(\ell)} \cdot \boldsymbol{A}_{i,j}^{(\ell)} \tag{21}$$

By marking the above value as $\boldsymbol{T}_{(i,j),(p,q)}^{(\ell)}$, we can get Lemma 2. $\qquad\square$

Based on Lemma 2, we further derive Theorem 2 as the pre-condition of the final Lemma 1.

**Theorem 2.** *By taking*

$$\boldsymbol{V}_{p,q}^{(\ell)} = \frac{\sum_{i,j} \boldsymbol{U}_{i,j}^{(\ell)} \cdot \boldsymbol{T}_{(i,j),(p,q)}^{(\ell)}}{\sum_{i,j} \boldsymbol{T}_{(i,j),(p,q)}^{(\ell)}}, \tag{22}$$

*we have*

$$\boldsymbol{u}^\top \nabla_{\boldsymbol{u}} L(\boldsymbol{w}, \boldsymbol{x}, \mathbf{1}) = \boldsymbol{v}^\top \nabla_{\boldsymbol{v}} L(\boldsymbol{w} \odot \mathbf{1}, \boldsymbol{x}). \tag{23}$$

*Proof.* According to Lemma 2,

$$\boldsymbol{u}^\top \nabla_{\boldsymbol{u}} L(\boldsymbol{w}, \boldsymbol{x}, \mathbf{1}) = \sum_\ell \sum_{i,j} \boldsymbol{U}_{i,j}^{(\ell)} \cdot \frac{\partial L(\boldsymbol{w}, \boldsymbol{x}, \mathbf{1})}{\partial \boldsymbol{U}_{i,j}^{(\ell)}}$$
$$= \sum_\ell \sum_{i,j} \boldsymbol{U}_{i,j}^{(\ell)} \cdot \sum_{p,q} \boldsymbol{T}_{(i,j),(p,q)}^{(\ell)}$$
$$= \sum_\ell \sum_{i,j} \sum_{p,q} \boldsymbol{U}_{i,j}^{(\ell)} \cdot \boldsymbol{T}_{(i,j),(p,q)}^{(\ell)}$$
$$= \sum_\ell \sum_{p,q} \sum_{i,j} \boldsymbol{U}_{i,j}^{(\ell)} \cdot \boldsymbol{T}_{(i,j),(p,q)}^{(\ell)} \tag{24}$$
$$= \sum_\ell \sum_{p,q} \frac{\sum_{i,j} \boldsymbol{U}_{i,j}^{(\ell)} \cdot \boldsymbol{T}_{(i,j),(p,q)}^{(\ell)}}{\sum_{i,j} \boldsymbol{T}_{(i,j),(p,q)}^{(\ell)}} \cdot \sum_{i,j} \boldsymbol{T}_{(i,j),(p,q)}^{(\ell)}$$
$$= \sum_\ell \sum_{p,q} \frac{\sum_{i,j} \boldsymbol{U}_{i,j}^{(\ell)} \cdot \boldsymbol{T}_{(i,j),(p,q)}^{(\ell)}}{\sum_{i,j} \boldsymbol{T}_{(i,j),(p,q)}^{(\ell)}} \cdot \frac{\partial L(\boldsymbol{w} \odot \mathbf{1}, \boldsymbol{x})}{\partial \boldsymbol{V}_{p,q}^{(\ell)}}.$$

By taking $\boldsymbol{V}_{p,q}^{(\ell)} = \frac{\sum_{i,j} \boldsymbol{U}_{i,j}^{(\ell)} \cdot \boldsymbol{T}_{(i,j),(p,q)}^{(\ell)}}{\sum_{i,j} \boldsymbol{T}_{(i,j),(p,q)}^{(\ell)}}$,

$$\boldsymbol{u}^\top \nabla_{\boldsymbol{u}} L(\boldsymbol{w}, \boldsymbol{x}, \mathbf{1}) = \sum_\ell \sum_{p,q} \boldsymbol{V}_{p,q}^{(\ell)} \cdot \frac{\partial L(\boldsymbol{w} \odot \mathbf{1}, \boldsymbol{x})}{\partial \boldsymbol{V}_{p,q}^{(\ell)}}$$
$$= \boldsymbol{v}^\top \nabla_{\boldsymbol{v}} L(\boldsymbol{w} \odot \mathbf{1}, \boldsymbol{x}). \tag{25}$$

Thus Theorem 2 is affirmed. $\qquad\square$

We now prove Lemma 1 equipped with Taylor Expansion technique and Theorem 2.

*Proof.* First, by adopting Taylor Expansions at $\boldsymbol{u} = \boldsymbol{0}$, we can get that:

$$L(\hat{\boldsymbol{w}}, \boldsymbol{x}, \mathbf{1} + \boldsymbol{u}) - L(\boldsymbol{w}, \boldsymbol{x}, \mathbf{1}) \approx L(\hat{\boldsymbol{w}}, \boldsymbol{x}, \mathbf{1}) + \boldsymbol{u}^\top \nabla_{\boldsymbol{u}} L(\hat{\boldsymbol{w}}, \boldsymbol{x}, \mathbf{1}) - L(\boldsymbol{w}, \boldsymbol{x}, \mathbf{1}). \tag{26}$$

Then, according to Theorem 2, the above equation can be rewritten as:

$$L(\hat{w}, x, 1 + u) - L(w, x, 1) \approx L(\hat{w} \odot 1, x) + v^\top \nabla_v L(\hat{w} \odot 1, x) - L(w, x, 1). \quad (27)$$

Again, by adopting Taylor Expansions at $v = 0$ for the right part of Eq. (27), we arrive at:

$$L(\hat{w}, x, 1 + u) - L(w, x, 1) \approx L(\hat{w} \odot (1 + v), x) - L(w, x, 1). \quad (28)$$

Finally, apply expectation on Eq. (28), and Lemma 1 is proved:

$$\begin{aligned}
\mathbb{E}_{x \sim \mathcal{D}_c}[L(\hat{w}, x, 1 + u(x)) - L(w, x, 1)] \\
\approx \mathbb{E}_{x \sim \mathcal{D}_c}[L(\hat{w} \odot (1 + v(x)), x) - L(w, x, 1)].
\end{aligned} \quad (29)$$

$\square$

With the Lemma 1 proved, we can easily derive Theorem 1 by the following transformation:

$$\begin{aligned}
\mathbb{E}_{x \sim \mathcal{D}_c}&[L(\hat{w}, x, 1 + u(x)) - L(w, x, 1)] \\
&\approx \mathbb{E}_{x \sim \mathcal{D}_c}[L(\hat{w} \odot (1 + v(x)), x) - L(w, x, 1)] \\
&= \mathbb{E}_{x \sim \mathcal{D}_c}[L(\hat{w} \odot (1 + v(x)), x, 1) - L(w, x, 1) + L(\hat{w}, x, 1) - L(\hat{w}, x, 1)] \\
&= \mathbb{E}_{x \sim \mathcal{D}_c}[L(\hat{w}, x, 1) - L(w, x, 1) + L(\hat{w} \odot (1 + v(x)), x, 1) - L(\hat{w}, x, 1)].
\end{aligned} \quad (30)$$

## C   EXPERIMENTS

### C.1   SUPPLEMENTARY EXPERIMENTS OF SEC. 3.1

To explore the upper limit of PTQ, we concern two key parts of the algorithm (weight quantization and activation one). As is generally known, QAT is a popular way to produce favor quantization results, thus we use QAT's learning mechanism to replace each of the two parts in PTQ and record outcomes in Table 7. In detail, about weight quantization, QAT's settings we used include the whole ImageNet, STE learning strategy and end-to-end training for 40 epochs while PTQ' settings only cover 1024 samples, training block by block. But we both keep the rounding-up-or-down parameter optimization space. About activation quantization, QAT' settings we used include the whole ImageNet, LSQ (Esser et al., 2019) scheme and end-to-end training for 5 epochs.

However, results in Table 7 are surprising that although the optimization space in weight is kept very restricted, leveraging the whole data to implement weight tuning can achieve a large accuracy boost. And the activation quantization step size might not be as important as weight. These findings indicate that exploring a quantization-friendly weight may be the fresh insight for the accurate PTQ, where this work dedicates to.

### C.2   SUPPLEMENTARY EXPERIMENTS OF SEC. 3.3

**Overfitting phenomenon.** To analyze the overfitting problem refered in Sec. 3.3, we have conducted experiments to compare the accuracy on test data and calibration data, respectively. The table below is an example of ResNet-18 W2A2. It can be seen that with extremely low-bit quantization, both Case 2 and Case 3 perform well on calibration data but on test data Case 2 performs worse than Case 3. This is a shred of clear evidence that Case 2 suffers a more severe overfitting problem.

| Activation setting | Weight setting | Accuracy |
|:---:|:---:|:---:|
| PTQ | PTQ | 46.64 |
| QAT | PTQ | 49.53 |
| PTQ | QAT | **57.49** |

Table 7: Ablation study of quantization settings on ResNet-18 W2A2.

| Methods | Test accuracy | Train accuracy |
|:---|:---:|:---:|
| Case 1 | 18.88 | 19.82 |
| Case 2 | 45.77 | 69.54 |
| Case 3 | 48.07 | 69.92 |
| QDROP | 51.14 | 66.11 |

Table 8: Train and test accuracy on ResNet-18 W2A2.

Rethinking Theorem 1, both Case 2 and Case 3 introduce the term (7-2) that implies flatness against the perturbation $v(x)$. However, Case 2 completely introduces the quantization noise $u(x)$ according to the calibration data. Thus the resulting flatness fits calibration data but does not generalize

well on test data. Case 3 drops partial of $u(x)$ and improves the possibility of flatness on test data instead. The two accuracy of QDROP also confirmed this phenomenon with more diverse directions of flatness. As for Case 1, its accuracy on both calibration data and test data is low. This is because that it does not introduce activation quantization during weight tuning and thus behaves worst without taking flatness term even on calibration data into account.

**Hessian information.** As Hessian information is known to be a metric to characterize flatness property (Keskar et al., 2016; Dong et al., 2019), we also calculate the top-1, top-5 Hessian eigenvalues ($\lambda_1, \lambda_5$) and the Hessian trace (Tr) among 3 cases and QDROP to better support our analysis. In table Table 9, QDROP has the smallest value of Hessian information, which matches with our theoretical framework and observations of loss landscape.

| Method | $\lambda_1$ | $\lambda_5$ | Tr |
|---|---|---|---|
| Case 1 | 14770 | 6746 | 122894 |
| Case 2 | 8423 | 4050 | 86287 |
| Case 3 | 8258 | 3821 | 84044 |
| QDROP | 6850 | 3044 | 66371 |

Table 9: Hessian information of the ResNet-18 W3A3 model. $\lambda_1$ represents for the top-1 Hessian eigenvalue, $\lambda_5$ for top-5 Hessian eigenvalues and Tr for Hessian trace.

### C.3 SUPPLEMENTARY EXPERIMENTS OF SEC. 4

To clarify the gap between AdaQuant and our method in classification task on 4-bit ResNet-18 and -50, we analyze the difference on settings between these two algorithms. There are two major discrepancies which would influence the final accuracy. One is the position of activation quantization nodes, the other is the FP32 accuracy of pretrain-models. As shown in Fig. 7, by inserting the quantizer like the right side one, AdaQuant would introduce two different quantizers for the same input with learned quantization parameters such as step size. We find this can bring $\sim 0.4\%$ upswings on ResNet-18 and -50 W4A4 and improve more on ultra low-bit with experiments of our algorithm. However, this way to insert quantization nodes is not practical in real deployment, because two different quantizers for the same input would make it impossible to follow the so-called Requantize procedure (Li et al., 2021b). As for the pretrain-models, they use 71.97% and 77.2%, higher than ours (71.06% and 77.0%), which can indeed improve the accuracy. Replacing our settings with the two adopted in AdaQuant, we finally reach 71.07% and 76.67%.

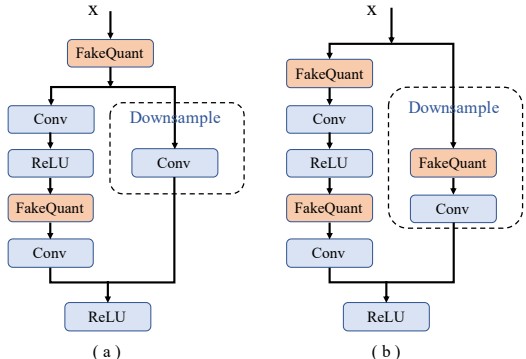

Figure 7: Different ways of inserting activation quantization nodes. Our method obeys the left side one while AdaQuant adopts the right side one.

| Method | Bits (W/A) | ResNet-18 | SWA$_{20}$ |
|---|---|---|---|
|  | FP32 | 71.06 | **71.50** |
| Min-Max* | 8/8 | 70.94 | **71.40** |
|  | 4/8 | 52.33 | **65.26** |
|  | 4/4 | 26.05 | **44.37** |
| OMSE* (Choukroun et al., 2019) | 32/4 | 64.15 | **65.99** |
|  | 4/4 | 38.32 | **55.86** |
| BRECQ†* (Li et al., 2021a) | 2/4 | 65.35 | **66.33** |
|  | 2/2 | 41.74 | **44.30** |

Table 10: Experiments between ResNet-18 and SWA$_{20}$ when adopting different PTQ methods. SWA$_{20}$ is acquired by finetuning ResNet-18 using SWA technique for 20 epochs.

### C.4 FLATNESS AND POST-TRAINING QUANTIZATION

While there are plenty of works exploring the correlation between flatness and generalization, the interaction between quantization and flatness has not been exploited much. In this paper, we first connect flatter quantized weight with activation quantization from PTQ view, as implied in Sec. 3.2. From this, we conjecture that flatness and quantization might be helpful to each other.

By leveraging some mechanisms devoting to produce a smoother loss surface for better generalization, such as (Izmailov et al., 2018), the improved FP32 model is obtained and used to validate the

performance on quantization compared with the naive one. In Table 10, model SWA$_{20}$ is enabled by applying the SWA technique (Izmailov et al., 2018) to ResNet-18 with 20 epochs fintuning process on the whole ImageNet. From the table, the observation is that the promotion on generalization can not fully represent the enhancement induced by SWA on quantization. With even lower bits thus larger noise, SWA$_{20}$ surpasses the naive one by a large margin. It also reveals that distinct FP32 models with analogous accuracy might contribute to surprisingly disparate outcomes after PTQ, particularly for those naive methods without any weight tuning.

In turn, there have been some studies that delve into robustness boost by applying quantization. (Fu et al., 2021) advocates that quantization can be properly leveraged to enhance DNNs' robustness, even beyond their full-precision counterparts. They propose a random bit training strategy to accomplish it, where our work illustrates the correlation with bit and perturbation in Appendix A.

## D    RELATED WORKS

**Post-training quantization.** Unlike QAT Esser et al. (2019); Li et al. (2019); Shen et al. (2021) where the quantized model is finetuned with full training dataset and over 100 epochs training, PTQ is much more faster. Rounding-to-nearest operation is known to be the direct and easy way for quantizing parameters or activations in PTQ. Although there is almost no accuracy drop when quantizing to 8-bit, lower bit quantization is yet a hard task and worth exploring. (Choukroun et al., 2019) transforms quantization to a Minimum Mean Squared Error problem both for weights and activations. (Nagel et al., 2019) equalizes weight ranges among channels thus be more favorable to per-layer quantization and employs bias correction to absorb the output error induced by quantization. However, such methods neglect the task loss thus lead to a sub-optimal. AdaRound (Nagel et al., 2020), which proposes to learn the rounding mechanism by reconstructing output layer by layer brings more opportunities for 4-bit quantization. Besides layer reconstruction, BRECQ (Li et al., 2021a) discusses more choices and advises to do block reconstruction with better accuracy at 2-bit weight quantization. Nonetheless, we argue that AdaRound and BRECQ isolate weight quantization and activation one theoretically and experimentally, which might be a key point of failures on extremely low-bit quantization. Recently, there is another trend of utilizing synthetic data for PTQ, which explicitly do backpropagation on the learned input tensor Cai et al. (2020); Zhang et al. (2021b); Li et al. (2021c).

**Flatness.** The idea of "flat" minima might date back to (Hochreiter & Schmidhuber, 1997), where the benefits are recognized in recent years, such as generalization (Jiang et al., 2019; Keskar et al., 2016) and adversarial training (Wu et al., 2020; Zheng et al., 2021). Some previous works (Izmailov et al., 2018; Foret et al., 2020) devote to improve the flatness of the trained weight for robustness under perturbation or distribution shift. Other ideas try to model flatness or sharpness formally by visualization of loss landscape or mathematical formulas. And for quantization, which could be viewed as some kind of noise, a flat model has been implied to be preferable, (Dong et al., 2019; Yang et al., 2019; Kadambi et al., 2020). Despite the natural fact that flatness helps with weight quantization, how does activation quantization involves with smoother loss surface has not been discussed deeply, particularly for post-training quantization. In this work, we introduce noise scheme by randomly dropping activation quantization and achieve a general flatness. Another paper (Fan et al., 2020) also utilizes randomness by adding noise to weight for the simulation of weight quantization. But they target at reducing the induced bias of Straight Through Estimation (STE) in QAT and also have different motivations and solve different problems from us.

## E    IMPLEMENTATION DETAILS

**Observation.** Here, we give the concrete implementation of experiments in Sec. 3.1. We actually consider three ways of introducing activation for Case 2, but we find the differences of outcomes among them are negligible thus employing one of them for clearer clarification.

**ImageNet.** We randomly extract 1024 training examples from ImageNet as calibration dataset based on the standard pre-process. Pretrain-models are downloaded from BRECQ's open source code. Hyper-parameters we keep it as BRECQ, such as batch size 32, learning rate for activation step size 4e-5, learning rate for weight tuning 1e-3, iterations 20000. Following BRECQ, we first fold batch normalization layer into convolution then reconstruct output block-wise to learn the weight

---

**Algorithm 2:** Implementations of three cases in Sec. 3.1

---

**Input:** Model with $K$ blocks.

**if** *Case 2* **then**
    **for** $k = 1$ *to* $K$ **do**
        ⌊ parameterize activation step size in this block ;

**for** $k = 1$ *to* $K$ **do**
    Tuning weight like (Li et al., 2021a) by reconstructing block output ;
    **if** *Case 3* **then**
    ⌊  ⌊ parameterize activation step size in this block ;

**if** *Case 1* **then**
    **for** $k = 1$ *to* $K$ **do**
        ⌊ parameterize activation step size in this block ;

**return** Quantized model ;

---

rounding policy and meanwhile use LSQ (Esser et al., 2019) to parameterize activation step size. For QDROP, we learn the weight and activation parameters together and use 50% rate to drop some activation quantization.

**Object detection.** Here, we also obey BRECQ' settings and use the same pretrain-models with 256 training samples taken from MS COCO dataset for calibration. Parameters about resolution is set to 800 (max size 1333) and 600 (max size 1000) for ResNets and MobileNetV2, respectively and batch size is set to 2 while others are the same with classification task. To be noted, we didn't quantize the head but applied block reconstruction to backbone and layer reconstruction to neck like BRECQ.

**GLUE benchmark and SQuAD.** The BERT fine-tuned models are taken from huggingface group (https://huggingface.co/). And we sampled 1024 examples from training set. We keep the maximum sequence length to be 128 for GLUE benchmark but maximum sequence length 384 with doc stride 128 for SQuAD1.1. Also, we quantize all the part in BERT as well as the internal structure of the attention module with only affirming the embedding weight to 8-bit. Other settings and hyper-parameters are chosen in the same way with ImageNet experiments.

**Baselines.** We run baseline methods from open-source codes, such as AdaQuant, BRECQ and Adaround. And we try our best to align some optimal settings like per-channel quantization for fair comparisons.

