# OpenReview forum: "QDrop: Randomly Dropping Quantization for Extremely Low-bit Post-Training Quantization"
_ICLR.cc/2022/Conference — ICLR 2022 Poster_

### Official Review · Reviewer_Y2p4 · 2021-10-20

**Correctness:** 2
**Technical Novelty And Significance:** 3
**Empirical Novelty And Significance:** 3
**Recommendation:** 8
**Confidence:** 4

**Main Review:**

Post-rebuttal
The response well address my concerns, and I would like to increase the score.
-----------------------------------------
Pros:
1. The idea of considering the activation quantization during reconstruction of PTQ is significant. As the authors demonstrate in Table that Case 1 underperforms Case 2 obviously, assigning more optimal weight calibration is highly demanded.

2. The observation of loss landscape flatness is also beneficial to the design of PTQ. Although the property of loss landscape of neural networks have been widely studied, the importance to PTQ remains to be explored.

3. The excremental results clearly show the presented methods outperform the SOTA especially in lightweight architectures such as MobileNet-V2, and the accuracy of 4-bit PTQ seems to be promising in large-scale dataset such as ImageNet.

Cons:
1. I think the core problem of this paper is the technical soundness of Eq. (7). I do not think the approximation in Eq. (14) and (15) is accurate Since the summation of the production in (19) is assumed to be exchangeable, is this always true? Otherwise, the form of Eq. (7) will be very complex.

2. The writing needs to be improved. The notations should be well defined before use. For example, what do $u(x)$ and $V(x)$ exactly mean? The authors only say  $1+u(x)$ represent the noise and this is not the formal definition.
Meanwhile, the connection between Eq. (5) and (6) should be specified.

3. What is the relationship between Corollary 1 and Eq (8)?

4. Figure 3 is not explained in the mainbody. What information can this figure convey? Meanwhile, why case 1 and case 3 seems to be similar?

5. The connection between flatness and randomly activation quantization dropping needs to be explained, i.e., the physical meaning of Eq. (9).

**Summary Of The Paper:**

This paper proposes the post-training quantization for extremely low-bit neural networks. By considering the activation quantization during reconstruction, the presented QDrop randomly drops the quantization of activations with higher loss flatness that adapts the activations with various activation bitwidths well. Experimental results have demonstrated the superiority of the presented method.

**Summary Of The Review:**

Please see the Cons in the main review.

---

> ### Author Response · Authors · 2021-11-18
> **Response to Reviewer Y2p4 Part II**
>
> - Q3: What is the relationship between Corollary 1 and Eq (8)?
>
>   A: For clarification, we give a general response ahead to summarize the connection between our overall theoretical analyses and the derived method. Here we make further explanation for this question. Corollary 1 says that "with activation quantization $\boldsymbol{u}(\boldsymbol{x})$, the trained model is flatter under weight perturbation $\boldsymbol{v}(\boldsymbol{x})$ for calibration data $\boldsymbol{x}$". $\boldsymbol{v}(\boldsymbol{x})$ (the noise on weight) is calculated from  $\boldsymbol{u}(\boldsymbol{x})$ (the noise on activation). Also note that one specific data corresponds to one $\boldsymbol{u(x)}$ thus one corresponding $\boldsymbol{v}(\boldsymbol{x})$ , then the flatness is dependent on input data and the introduced quantization noise (i.e. $\boldsymbol{u(x)}$).
>
>   Eq. (8) analyzes the actual situation of  $\boldsymbol{u(x)}$ for the 3 Cases during tuning. Case 1 ignores activations quantization and thus has the worst flatness and poor performance on both calibration and test data. Case 2 considers total activation perturbation and will produce a flat model under calibration data. But Case 3 drops some activation noise and will get different weight perturbation. This contributes to the flatness under other directions which are not restricted to calibration data. Thus although Case 2 and 3 perform similarly on calibration data, Case 3 behaves better on test data. We have revised Corollary 1 with the concrete noise notation to make it clearer.
>
> - Q4: Figure 3 is not explained in the main body. What information can this figure convey? Meanwhile, why case 1 and case 3 seems to be similar?
>
>   A: Thanks for pointing that out. About Fig. 3, we leverage [1] to draw the loss landscape of the quantized model (ResNet-18 W3A3) on ImageNet test data. Our aim is to provide some visual evidence that QDrop leads to a flatter model, where QDrop in Fig. 3 shows a smoother loss landscape than Case 3 (the winner among 3 Cases) and Case 1 (the naive method described in [2]).
>
>   As for the similarity of Case 1 and Case 3 in Fig. 3, it is because we visualize a relatively large area around the minimum. When we zoom in the local area around the minimum and can find that Case 3 is obviously flatter than Case 1 on test data (see the updated Fig. 3 in the revision). Also, as advised by Reviewer iK8W in Q1, we calculate some Hessian information (top-1, top-5 Hessian eigenvalues ($\lambda_1$, $\lambda_5$), and the Hessian trace ($Tr$)) to further validate this point. Here are the results. It can be seen that Case 3 has smaller Hessian eigenvalues and traces than Case 1. This can also prove that Case 3 is flatter than Case 1 from a quantitative perspective.
>
>   | Method | $\lambda_1$ | $\lambda_5$ | $Tr$ |
>   | --- | --- | --- | --- |
>   | Case 1 | 14770 | 6746 | 122894 |
>   | Case 2 | 8423 | 4050 | 86287 |
>   | Case 3 | 8258 | 3821 | 84044 |
>   | QDrop | 6850 | 3044 | 66371 |
>
> - Q5: The connection between flatness and randomly activation quantization dropping needs to be explained, i.e., the physical meaning of Eq. (9).
>
>   A: Thanks for your advice and we have revised our paper to explain it more clearly. The noise on weight $\boldsymbol{v}(\boldsymbol{x})$ is calculated by using $\boldsymbol{u}(\boldsymbol{x})$.  And they correlate with input data (calibration one during tuning) and how we quantize activation. By randomly dropping activation quantization each forward pass, we can get different $\boldsymbol{u}(\boldsymbol{x})$ and this can lead to more kinds of $\boldsymbol{v}(\boldsymbol{x})$. With diverse $\boldsymbol{v}(\boldsymbol{x})$ , the flatness will not be limited to perturbation related to calibration data and can be extended to test one. This explains why our QDrop performs better than the other cases.
>
> [1] Li H, Xu Z, Taylor G, et al. Visualizing the loss landscape of neural nets[J]. arXiv preprint arXiv:1712.09913, 2017.
>
> [2] Li Y, Gong R, Tan X, et al. Brecq: Pushing the limit of post-training quantization by block reconstruction[J]. arXiv preprint arXiv:2102.05426, 2021.

---

> ### Author Response · Authors · 2021-11-18
> **Response to Reviewer Y2p4 Part I**
>
> We would like to thank you for your thoughtful review and insight on this paper. The detailed response is listed below and we hope we address your concerns.
>
> - Q1: "I think the core problem of this paper is the technical soundness of Eq. (7). I do not think the approximation in Eq. (14) and (15) is accurate Since the summation of the production in (19) is assumed to be exchangeable, is this always true? Otherwise, the form of Eq. (7) will be very complex."
>
>   A: We'd like to demonstrate the technical soundness of Eq. (7) by considering the cases of fully connected and convolutional networks separately.
>
>   - For the case of the fully connected layer, we prove with Eq. (5) in the main body of the paper. The derivation is natural to understand and we also add more detailed proofs about this situation in Appendix B.1 to make it clearer in the revision.
>
>   - For the convolutional one, we put it in Appendix B.2.  In the original manuscript, we used an independence hypothesis and arrived at the approximate equation in Eq. (19). We think the proposed concern is about the independence hypothesis. Thanks a lot for pointing that out. We reconsidered the problem and have demonstrated it in a similar but easier way, which does not need this hypothesis. Here, we temporarily omit $\boldsymbol{\boldsymbol{x}}$ in the notation below for simplicity.  In summary, we construct two networks that insert noise on activation ($\boldsymbol{u}$) and weight ($\boldsymbol{v}$), respectively, and calculate the first-order derivative to $\boldsymbol{u}$ and $\boldsymbol{v}$ at $\boldsymbol{0}$ as clarified in the original version. Then we propose and prove that there exists some $\boldsymbol{v}$ that satisfies the following equation and thus the noise on activation can be transformed into weight.
>
>     $$ {\boldsymbol{u}}^{\top}\nabla_{\boldsymbol{u}}L(\boldsymbol{w}, \boldsymbol{1})={\boldsymbol{v}}^{\top}\nabla_{\boldsymbol{v}}L(\boldsymbol{w}\odot{\boldsymbol{1}})$$
>
>     In the original paper, we work on the expectation form and decouple it with the independence hypothesis, which is not accurate as you pointed out. But here, we find that with $\boldsymbol{V}\_{p,q}^{(\ell)}=\frac{\sum\_{i,j}\boldsymbol{U}\_{i,j}^{(\ell)}\cdot \boldsymbol{T}\_{(i,j),(p,q)}^{(\ell)}}{\sum\_{i,j}\boldsymbol{T}\_{(i,j),(p,q)}^{(\ell)}}$, the above equation holds. The hypothesis is no longer needed. We list the concrete and detailed proof in Theorem 2 in Appendix B.2 of the revision. Finally, leveraging Taylor expansion, we can reach Lemma 1 or Eq. (7) as before.
>
>   We have reorganized the overall proof of Lemma 1 and Theorem 1 in Appendix B for clearer clarification.
>
> - Q2: "The writing needs to be improved. The notations should be well defined before use. For example, what do $\boldsymbol{u(x)}$ and $\boldsymbol{V(x)}$ exactly mean? The authors only say $1+\boldsymbol{u(x)}$ represent the noise and this is not the formal definition. Meanwhile, the connection between Eq.(5) and (6) should be specified."
>
>   A: Thanks for the useful advice.  We add some definitions about $\boldsymbol{u(x)}$ and $\boldsymbol{V(x)}$ before use in the revision.  $\boldsymbol{u(x)}$ is the noise on activation and is defined by the fixed-point integer and rounding error. The illustration of the physical meaning of $\boldsymbol{u(x)}$ is put in Appendix A.  $\boldsymbol{V(x)}$ means perturbation on the weight and is derived from $\boldsymbol{u(x)}$ where we give the form in FC layer in the main body and extended the form to convolutional structure in Appendix B.2.
>
>   Also, we specify the connection between Eq.(5) and (6) in detail in Appendix B.1. In short, Eq. (5) shows that during the forward pass, we can transform noise on activation into the weight for each layer. Thus the optimization objective with noise on activation can be transformed into the optimization objective with noise on weight, i.e., Eq. (6).

---

### Official Review · Reviewer_5Zj1 · 2021-11-02

**Correctness:** 4
**Technical Novelty And Significance:** 4
**Empirical Novelty And Significance:** 4
**Recommendation:** 8
**Confidence:** 5

**Main Review:**

This paper is well-organized, and its goal is well-oriented by providing a good motivational experiment. Theoretical analyses are well-balanced with empirical observations to offer a valuable insight that the flatness of the model is crucial to the extremely low-bit PTQ. However, there seem to be some glitches in formulating and presenting a theoretical framework. The novelty of the QDrop algorithm itself is moderate and simple, but the intuition behind it is critical. Experimental results look promising. Here are a few concerns and questions:

1. An essential basis of the given theoretical framework is an assumption that the activation quantization could be modeled as injecting a multiplicative form of perturbation, say a^ = a + e = a + au = a(1+u) = as, where s is a scaling factor. Is this assumption reasonable in an actual situation? If this assumption is not available, is the conclusion still valid? Is it possible to be represented by any generalized form of perturbation?

2. An operator used in the vector/matrix multiplication should be defined. Is it element-wise multiplication or ordinary matrix-vector multiplication? Does it have a commutative property? For example, in Eq.(5) and the equation above Eq.(5), could you check the dimension of vectors and matrices? Could you provide any proof of Eq.(5)? It seems to be crucial to prove the following Lemma 1 and Theorem 1. The proof of Sec. 4.2 provided in Appendix A is a bit confusing to the reviewer. Could you clean up the critical statements by setting up Corollary or Lemma ones?

3. Several vague and confusing terminologies were used in Corollary 1. For example, the trained quantized weights are usually(?) flatter on some(?) directions and more robust under some(?) perturbations for calibration data. Is there any other case that the trained quantized weights are not flattering? Again, technical terms should be used to give some intuitions.

4. In Section 4.3, the authors claim that it is highly possible(??) that Case 2 causes overfitting due to the mismatch on calibration and test data. However, Case 2 and Case 3 seem to perform activation quantization based on Eq.(7) using the same calibration data. Could you prove that the relatively low performance of Case 2 compared with Case 3 is due to the overfitting? Their performance gap is just 2% in most of the models in Table 1. If overfitting occurs during the weight tuning process, it seems to be suitable to arise in Case 1, which does not perform activation quantization. There should be a supplementary explanation for this by comparing the results before and after weight tuning.

5. When equivalent compression levels are applied by quantization, it seems more evident that the proposed method can improve generalization performance while preventing expected compression losses. Could you evaluate the average precision(bit-width) of QDrop in the experiments? If the average precision is not critical to this case, could you explain the reason?

6. It is generally known that the performance of QAT is relatively good compared to PTQ. Are some ablation studies necessary to provide supplementary experiments with a QAT setting in Appendix B?

Minor Comments:
1. (Singh et al., 2019) is wrongly cited on page 2. Please refer to (Nagel et al., 2019) and (Banner et al., 2019) instead.
2. There are several typos: (1) are->is on page 2, (2) for each architecture, we ~~ on page 7, (3) origin->original on page 9, (4) Fig. 5.3 should be replaced with Table 6 on page 9, (5) Table ?? on page 15.
3. In Table 2 and Table 6, please unify the same terms, say No Drop vs. No QDROP. Does it correspond to Case 2?
4. The caption title is missing in the table on page 15. In the table, what is SWA_20? and 32/4 is right in OMSE? It seems that the order of rows in OMSE should be sorted in this case.


**Summary Of The Paper:**

This paper aims to analyze how activation quantization affects the PTQ process and provides some theoretical analysis and experimental results on the effects of activation quantization. The authors argue that previous studies only model the weight quantization as perturbation while ignoring activation quantization, causing a sub-optimal solution by missing out on the main factor in the performance degradation of quantized models in low-precision PTQ environments. Based on the empirical and theoretical analyses, this paper proposes QDrop to pursue flatness and demonstrate that partial activation quantization is more beneficial.

Contributions:
1. Observed the benefits of activation quantization in low precision PTQ.
2. Conducted theoretical studies on how activation quantization affects weight tuning.
3. Presented QDrop by showing that both integrating activation quantization into PTQ reconstruction and dropping partial activation quantization may help the flatness of the model, which is vital to the final accuracy.
4. Established a new SOTA for W2/A2 PTQ with QDrop.

**Summary Of The Review:**

To sum up, the paper is well-written and generally interesting even though there are a few glitches. Therefore, I may change my rating according to the authors' rebuttal to resolve the above concerns and questions.

***** Post-Rebuttal Comments *****
I appreciate the authors for their detailed response. The newly updated manuscript addresses most of my concerns. Therefore, I am happy to raise my rating up and genuinely recommend this paper to be accepted.

---

> ### Author Response · Authors · 2021-11-09
> **Question about your review**
>
> Thanks for your detailed comments.
>
> Regarding your question 5, we had some trouble understanding it. We don't use mix-precision and adopt unified bit-width in our experiments. So we want to know what the *average bit-width* and *equivalent compression levels* refer to. Could you please further clarify this question? It would be helpful for our response. Looking forward to your reply! Thanks!

---

> > ### Comment · Reviewer_5Zj1 · 2021-11-11
> > **Answer about your question**
> >
> > Thanks for your question to prepare your rebuttal. Sorry for the confusion. Even though your scheme is not a mixed-precision quantization, the proposed method employs a quantization dropping policy which drops about half of the unified bit-width quantization. Does it use FP32 in case of the dropped situation? If it is the case, there would be an average bit-width in the process of overall quantization/compression. But I am not sure whether the average bit-width or how much the model is compressed is critical in your targeted problem. I wish this may resolve the confusion. If you still have trouble understanding the question, you may ignore the response to question 5.

---

> > > ### Author Response · Authors · 2021-11-11
> > > **Reply to your answer**
> > >
> > > Thanks for your timely reply and clear explanation! We have understood the question. We are preparing the detailed reply for all the questions and will post them as soon as possible.

---

> ### Author Response · Authors · 2021-11-18
> **Response to Reviewer 5Zj1 IV**
>
> * Q5: "When equivalent compression levels are applied by quantization, it seems more evident that the proposed method can improve generalization performance while preventing expected compression losses. Could you evaluate the average precision(bit-width) of QDrop in the experiments? If the average precision is not critical to this case, could you explain the reason?"
>
>   A: In fact, we only drop some activation quantization during the calibration tuning process to optimize the model. During the test phase or the final deployment, all the quantization is still kept. So our method will not hurt the compression rate. We have clarified this point more evidently in the Implementation Details part of the revision.
>
> * Q6: "It is generally known that the performance of QAT is relatively good compared to PTQ. Are some ablation studies necessary to provide supplementary experiments with a QAT setting in Appendix B?"
>
>   A: These supplementary experiments in Appendix B (Appendix C.1 in the revision) aim to explore the upper limit of PTQ. As is generally known that QAT is relatively good, we replace each part of PTQ (weight quantization, activation quantization) with QAT settings (the whole ImageNet data and larger training epochs), respectively. We use "No Drop" as the baseline and find that with favorable quantized weight (the last row of Table 8), performance can be largely enhanced. This experiment also inspires us to pursue more optimal weight. And this paper theoretically and empirically finds that producing the optimal model from a flatness perspective is effective for extremely low-bit PTQ.
>
> * Q7: "In Table 2 and Table 6, please unify the same terms, say No Drop vs. No QDROP. Does it correspond to Case 2?"
>
>   A: Thanks for pointing that out. We have unified them to "No Drop". "No Drop" means setting the dropping probability to zero and corresponds to only removing QDrop. "No Drop" has the same principle and tuning pipeline as Case 2 with only one small difference of implementation detail. In "No Drop", we learn the step size for the activation quantizer to fairly compare with QDrop. In Case 2, we set the step size by collecting activation distribution for fair comparisons among 3 Cases. We can find that the little difference has little influence on accuracy and the conclusion is consistent that QDrop optimizes a flatter model.
>
> * Q8: "The caption title is missing in the table on page 15. In the table, what is SWA_20? and 32/4 is right in OMSE? It seems that the order of rows in OMSE should be sorted in this case."
>
>   A:  Thanks for pointing that out. We finetune our ResNet-18 with SWA [3] technique for 20 epochs and then get the model SWA$\_{20}$. SWA technique is known to be a useful method to find much flatter local minimal than SGD. The experimental results of SWA$\_{20}$ further prove our finding that the flatness is helpful for PTQ. And we have added the caption title and sorted the order of rows in OMSE.
>
> * Q9: The other typos pointed out in minor comments.
>
>   A: Thanks for your suggestions. We have revised them and marked them in red.
>
> [1] Jiang Y, Neyshabur B, Mobahi H, et al. Fantastic generalization measures and where to find them[J]. arXiv preprint arXiv:1912.02178, 2019.
>
> [2] Keskar N S, Mudigere D, Nocedal J, et al. On large-batch training for deep learning: Generalization gap and sharp minima[J]. arXiv preprint arXiv:1609.04836, 2016.
>
> [3] Izmailov P, Podoprikhin D, Garipov T, et al. Averaging weights leads to wider optima and better generalization[J]. arXiv preprint arXiv:1803.05407, 2018.

---

> ### Author Response · Authors · 2021-11-18
> **Response to Reviewer 5Zj1 III**
>
> * Q3: "Several vague and confusing terminologies were used in Corollary 1. For example, the trained quantized weights are usually(?) flatter on some(?) directions and more robust under some(?) perturbations for calibration data. Is there any other case that the trained quantized weights are not flattering? Again, technical terms should be used to give some intuitions."
>   A: Thanks for the suggestion. We have revised Corollary 1 to make the illustration more formal:
>
>   > "On calibration data $\boldsymbol{x}$, with activation quantization noise $\boldsymbol{u(x)}$, there exists the corresponding weight perturbation $\boldsymbol{v(x)}$ which satisfies that the trained quantized model is flatter under the perturbation of $\boldsymbol{v(\boldsymbol{x})}$."
>
>   The corollary means that we can obtain a flat model under the perturbation of $\boldsymbol{v(x)}$ when introducing activation quantization noise of $\boldsymbol{u(x)}$. As stated in the previous question, any activation noise $\boldsymbol{u(\boldsymbol{x})}$ can be transformed into a specific relevant weight noise $\boldsymbol{v(x)}$. Thus the perturbation that the model behaves flat against is related to the quantization noise. This explains why the different ways to introduce activation quantization influence the final performance. Inspired by this, we further design the QDrop to cover diverse $\boldsymbol{u(x)}$, and thus achieve the flatness across various perturbations ($\boldsymbol{v(x)}$) that helps improve the accuracy.
>
> * Q4: "In Section 4.3, the authors claim that it is highly possible(??) that Case 2 causes overfitting due to the mismatch on calibration and test data. However, Case 2 and Case 3 seem to perform activation quantization based on Eq.(7) using the same calibration data. Could you prove that the relatively low performance of Case 2 compared with Case 3 is due to the overfitting? Their performance gap is just 2\% in most of the models in Table 1. If overfitting occurs during the weight tuning process, it seems to be suitable to arise in Case 1, which does not perform activation quantization. There should be a supplementary explanation for this by comparing the results before and after weight tuning."
>
>   A: To analyze the overfitting problem, we have conducted experiments to compare the accuracy on test data and calibration data, respectively. The table below is an example of ResNet-18 W2A2. It can be seen that with extremely low-bit quantization, both Case 2 and Case 3 perform well on calibration data but on test data Case 2 performs worse than Case 3. This is a shred of clear evidence that Case 2 suffers a more severe overfitting problem. Rethinking Eq. (7), both Case 2 and Case 3 introduce the term (7-2) that implies flatness against the perturbation $\boldsymbol{v(x)}$. However, Case 2 completely introduces the quantization noise $\boldsymbol{u(x)}$ according to the calibration data. Thus the resulting flatness fits calibration data but does not generalize well on test data. Case 3 drops partial of $\boldsymbol{u}(\boldsymbol{x})$ and improves the possibility of flatness on test data instead. As for Case 1, its accuracy on both calibration data and test data is low. This is because that it does not introduce activation quantization during weight tuning and thus behaves worst without taking flatness term even on calibration data into account. We have added this discussion and experimental results in Appendix C.3.
>
>   | Method | Accuracy on Test Data | Accuracy on Calibration data |
> | ------ | --------------------- | ---------------------------- |
> | Case 1 | 31.26                 | 34.67                        |
> | Case 2 | 50.86                 | 70.12                        |
> | Case 3 | 52.83                 | 70.61                        |
> | QDrop  | 54.72                 | 66.50                        |

---

> ### Author Response · Authors · 2021-11-18
> **Response to Reviewer 5Zj1 II**
>
> - Q2: "An operator used in the vector/matrix multiplication should be defined. Is it element-wise multiplication or ordinary matrix-vector multiplication? Does it have a commutative property? For example, in Eq.(5) and the equation above Eq.(5), could you check the dimension of vectors and matrices? Could you provide any proof of Eq.(5)? It seems to be crucial to prove the following Lemma 1 and Theorem 1. The proof of Sec. 4.2 provided in Appendix A is a bit confusing to the reviewer. Could you clean up the critical statements by setting up Corollary or Lemma ones?"
>
> - A:  Thanks for the useful advice. We have revised the paper including defining operators of element-wise and matrix-vector multiplication, correcting the dimension fault in Eq.(5) where we wrote the matrix to its transpose and reorganizing the proofs in Appendix B.
>
>   As for the left questions, we explain as follows:
>
>   - **Detailed proof of Eq. (5):**
>
>     For Eq.(5), we consider the matrix-vector multiplication $\boldsymbol{W}\boldsymbol{a}$ and try to transform the noise on activation ($\boldsymbol{u(\boldsymbol{x})}$) into perturbation on weight ($\boldsymbol{V(\boldsymbol{x})}$).
>
>     $\boldsymbol{W}(\boldsymbol{a} \odot \begin{bmatrix} 1+\boldsymbol{u}_1\boldsymbol{(x)}\\\\1+\boldsymbol{u}_2\boldsymbol{(x)}\\\\ ...\\\\ 1+\boldsymbol{u}_n\boldsymbol{(x)}\end{bmatrix})=(\boldsymbol{W} \odot \begin{bmatrix} 1+\boldsymbol{u}_1\boldsymbol{(x)} & 1+\boldsymbol{u}_2\boldsymbol{(x)} & ... & 1+\boldsymbol{u}_n\boldsymbol{(x)} \\\\ 1+\boldsymbol{u}_1\boldsymbol{(x)}& 1+\boldsymbol{u}_2\boldsymbol{(x)}& ... & 1+\boldsymbol{u}_n\boldsymbol{(x)}\\\\ ...\\\\1+\boldsymbol{u}_1\boldsymbol{(x)}& 1+\boldsymbol{u}_2\boldsymbol{(x)}& ... & 1+\boldsymbol{u}_n\boldsymbol{(x)}\end{bmatrix})\boldsymbol{a}.$
>
>     For each sample $\boldsymbol{\boldsymbol{x}}$, we demonstrate it by looking at each element in the output and temporarily omit $\boldsymbol{\boldsymbol{x}}$ in the notation below for simplicity.
>
>     $\boldsymbol{z}\_i^{(\ell+1)}=\sum_j\boldsymbol{W}\_{i,j}^{(\ell)}\cdot (1+\boldsymbol{u}\_j^{(\ell)})\cdot \boldsymbol{a}\_j^{(\ell)}=\sum_j(1+\boldsymbol{u}\_{j}^{(\ell)})\cdot \boldsymbol{W}\_{i,j}^{(\ell)}\cdot \boldsymbol{a}\_j^{(\ell)}$
>
>     By taking $\boldsymbol{V}_{i,j}^{(\ell)}=\boldsymbol{u}_j^{(\ell)}$ for each input sample, we can achieve Eq. (5), i.e., $\boldsymbol{W}(\boldsymbol{a}\odot(\boldsymbol{1}+\boldsymbol{u}(\boldsymbol{x}))) = (\boldsymbol{W}\odot(\boldsymbol{1}+\boldsymbol{V}(\boldsymbol{x})))\boldsymbol{a}$ in short expression.
>
>     We have added this detailed proof in Appendix B.1.
>
>   - **Further clarification for proof of Lemma 1 and Theorem 1 in Appendix (** clean up the critical statements by setting up Corollary or Lemma ones **)**:
>
>     As for the proof of Lemma 1 and Theorem 1 in the Appendix, we reorganize it to make the paper clearer. It can be found in Appendix B in the revision. Here, we give a brief summary for a better understanding.
>
>     We prove Lemma 1 by considering the situation of fully connected (Appendix B.1) and convolutional networks (Appendix B.2) separately.
>
>     (1) For FC structure, according to Eq. (5) and the above analysis, during forward pass, we can make this transformation each layer. Thus, the two optimization objective in Lemma 1 satisfy the Eq. (6).
>
>     (2) For convolutional structure, we prove that transforming the noise on activation into weight can also be achieved. We first introduce two networks $\mathcal{G}\_1$, $\mathcal{G}\_2$ in Eq. (15) for better clarification. $\mathcal{G}\_1$ matches with adding noise ($\boldsymbol{u}(\boldsymbol{x})$) on activation while $\mathcal{G}\_2$ matches with adding noise ($\boldsymbol{v}(\boldsymbol{x})$) on weight. We find there are common parts in their first-order derivative to its noise as described in Lemma 2. With this, Theorem 2 can be derived, which presents that, for a specific $\boldsymbol{u}(\boldsymbol{x})$, we can calculate the corresponding $\boldsymbol{v}(\boldsymbol{x})$ such that ${\boldsymbol{u}}^{\top}\nabla_{\boldsymbol{u}}L(\boldsymbol{w}, \boldsymbol{x},\boldsymbol{1})={\boldsymbol{v}}^{\top}\nabla_{\boldsymbol{v}}L(\boldsymbol{w}\odot{\boldsymbol{1}},\boldsymbol{x})$. Finally, with this theorem and Taylor first-order expansion technique, the Eq. (6) satisfies and the Lemma 1 is proved.
>
>     Then, by interpolating the $L(\hat{\boldsymbol{w}}, \boldsymbol{x})$ in to the right side of Eq. (6) in Lemma 1, Theorem 1 holds.

---

> ### Author Response · Authors · 2021-11-18
> **Response to Reviewer 5Zj1 I**
>
> We would like to sincerely thank the reviewer for providing a constructive review and detailed comments. We hope our reply can address the questions.
>
> - Q1: "An essential basis of the given theoretical framework is an assumption that the activation quantization could be modeled as injecting a multiplicative form of perturbation, say a^ = a + e = a + au = a(1+u) = as, where s is a scaling factor. Is this assumption reasonable in an actual situation? If this assumption is not available, is the conclusion still valid? Is it possible to be represented by any generalized form of perturbation?"
>
>   A: The multiplicative form is reasonable and in line with the actual situation under the quantization background ($\hat{a}=\lfloor\frac{a}{s}\rceil\cdot s$). (here $s$ is the quantization step size and is not the same $s$ as the reviewer mentioned.)
>
>   First, this form is not an assumption and is equivalently transformed from the generalized noise under the quantization setting. So it will not hurt the effectiveness of the conclusion.
>
>   Second, the traditional noise form ($\hat{a} = a + e$) does not decouple the noise from the range of activations. The range of noise $e$ will vary when the range of activations change, making it complicated to analyze across the whole network in a unified form. Some existing papers (e.g., [1], [2]) also indicate the problem that the additive noise doesn't take the magnitude of parameters into account and suggest a multiplicative form one. To be specific, they claim that
>
>   > Perturbing the parameters without taking their magnitude into account can cause many of them to switch signs. Therefore, one cannot apply large perturbations to the model without changing the loss significantly. One possible modification to improve the perturbations is to choose the perturbation magnitude based on the magnitude of the parameter. In that case, it is guaranteed that if the magnitude of perturbation is less than the magnitude of the parameter, then the sign of the parameter does not change.
>
>   However, the multiplicative form avoids the problem. With the form $\hat{a}=(1+u)\cdot{a}$, $u$ can be formulated as $u=\frac{\hat{a}}{a}-1$. In the quantization background, we can denote the rounding error as $c$ and the fix-point integer value corresponding to $a$ as $\bar{a}$, then $u=\frac{\bar{a}\cdot s}{(\bar{a}+c) \cdot s}-1=\frac{\bar{a}}{\bar{a}+c}-1=\frac{-c}{\bar{a}+c}$. From this formulation, we can find that the range of $u$ is not related to the activation range or step size $s$ and is only influenced by the rounding error and the bit-width.
>
>   Thus the multiplicative form has its actual meaning in the quantization background, eliminates the influence of activation range on noise format, and meanwhile can totally represent the generalized noise form under the quantization setting. We have revised the paper to make the illustration of $u$ clearer in Appendix A.

---

### Official Review · Reviewer_gKCS · 2021-11-02

**Correctness:** 3
**Technical Novelty And Significance:** 2
**Empirical Novelty And Significance:** 2
**Recommendation:** 6
**Confidence:** 3

**Main Review:**

Strengths:

1. Do exploration on activation quantization and transforming the activation quantization into weight perturbations is a good idea.

2. Experimental results on many tasks and models show the QDROP's advantages.
Ablation studies also support the analysis.

3. This paper is generally well organized.

Weakness:

1. The terms of 'flatness' and 'sharpness' are not well defined, it might make readers confused on these and hard to follow.
Can the authors provide more detailed explanations on these two terms?

2. About Figure 3, the reviewer did not find any related explanation about these figures.

3. The reviewer expected a discussion on how does activation quantization involves with smoother loss surface,
while it looks missing in this work.

Minor issues:

1. Impact of dropping probability on ImageNet is performed on two network structures.
Can the authors provide more experiments on other networks? That will be more convincing.

2. On page 3, it should be H^{w} in the line below the equation (3)?

**Summary Of The Paper:**

The authors proposed a random dropping quantization method at the post-training stage
to achieve a low bit quantization network.
By observing the performance on partial activation quantization,
the authors analyze the influence of incorporating activation quantization into weight tuning.
Experimental results show that their QDROP methods have benefits on several scenarios, such as detection and NLP tasks.

**Summary Of The Review:**

Overall, I think this will be an instructive work if the authors can tackle the reviewer's concerns.

---

> ### Author Response · Authors · 2021-11-18
> **Response to Reviewer gKCS Part II**
>
> - Q4: "Impact of dropping probability on ImageNet is performed on two network structures. Can the authors provide more experiments on other networks? That will be more convincing."
>
>   A: Thanks for your good advice. Besides the traditional large ResNet-50 model and light-weight MobileNetV2, we further conduct experiments on two NAS SOTA models: RegNet-600MF and MNasNet-2.0. Here are the results, where taking dropping probability as $0.5$ still behaves best overall.
>
>   | Model        | Bits (W/A) | $p=0.0$ | $p=0.25$ | $p=0.5$ | $p=0.75$ | $p=1.0$ |
>   | ------------ | ---------- | ------- | -------- | ------- | -------- | ------- |
>   | RegNet-600MF | 2/3        | 55.38   | 57.56    | **58.40**  | 57.95    | 26.42   |
>   | MNasNet-2.0  | 2/3        | 47.61   | 55.33    | **55.80**   | 53.88    | 0.922   |
>
> - Q5: "On page 3, it should be H^{w} in the line below the equation (3)?"
>
>   A: Thanks for pointing that out. We have revised it and marked it in red.
>
> [1] Keskar N S, Mudigere D, Nocedal J, et al. On large-batch training for deep learning: Generalization gap and sharp minima[J]. arXiv preprint arXiv:1609.04836, 2016.
>
> [2] Neyshabur B, Bhojanapalli S, McAllester D, et al. Exploring generalization in deep learning[J]. arXiv preprint arXiv:1706.08947, 2017.
>
> [3] Jiang Y, Neyshabur B, Mobahi H, et al. Fantastic generalization measures and where to find them[J]. arXiv preprint arXiv:1912.02178, 2019.
>
> [4] Li H, Xu Z, Taylor G, et al. Visualizing the loss landscape of neural nets[J]. arXiv preprint arXiv:1712.09913, 2017.

---

> ### Author Response · Authors · 2021-11-18
> **Response to Reviewer gKCS Part I**
>
> We thank the reviewer for the constructive comments and feedback. Hope the following reply helps address the concerns.
>
> - Q1: "The terms of 'flatness' and 'sharpness' are not well defined, it might make readers confused on these and hard to follow. Can the authors provide more detailed explanations on these two terms?"
>
>   A:  In the original submission, we made some basic discussions on the two concepts after Theorem 1 and now we will give a more detailed explanation to make it clearer.
>
>   As illustrated in [1], intuitively, flat minimum means relatively small loss change under perturbation in the parameter space, otherwise, the minimum is sharp. In this paper, we follow the notion of flatness defined in [2], which considers loss change from the perspective of statistical expectation. And as [2] and [3] refer to, we consider the magnitude of the perturbation with respect to the magnitude of parameters and take the formulation below:
>
>   $\mathbb E_{\boldsymbol{v\sim\mathcal{D}}}  [L(f_{\boldsymbol{w}\odot (\boldsymbol{1}+\boldsymbol{v})})-L(f_{\boldsymbol{w}})]$
>
>   where each element of $\boldsymbol{v}$ is a random variable sampled from a noise distribution $\mathcal{D}$ and $L$ represents for optimization objective on the training set.
>
>   We have revised our paper to make this point clearer.
>
> - Q2: "About Fig. 3, the reviewer did not find any related explanation about these figures."
>
>   A: Thanks for pointing this out. We are really sorry we forget this part. In this figure, we leverage [4] to plot the loss landscape of the different Cases and our method, where we show QDrop reaches a flatter minimum.
>
>   Also, Reviewer iK8W suggests in Q1 that we can also validate this point via Hessian metrics. And we find that Hessian spectral results can also support that QDrop produces a  flatter quantized model, which further quantitatively proves the effectiveness of our method.
>
> - Q3: "The reviewer expected a discussion on how does activation quantization involves with smoother loss surface, while it looks missing in this work."
>
>   A: In the theoretical analysis, we find that quantization noise $\boldsymbol{u(x)}$ on activation can be transformed into perturbation $\boldsymbol{v(x)}$ on weight (Eq. (5)). Thus introducing activation quantization brings an extra term in the optimization objective (Theorem 1), i.e.,
>
>   $\mathbb{E}_{\boldsymbol{x}\sim\mathcal{D}_c}[(L(\hat{\boldsymbol{w}}\odot(\boldsymbol{1}+\boldsymbol{v(x)},\boldsymbol{x}) - L(\hat{\boldsymbol{w}}, \boldsymbol{x})]$
>
>   The above equation stands for loss change with jitters on the quantized weight, i.e. the flatness of model. By considering this loss term during optimization, we can reach a smoother loss surface under perturbation $\boldsymbol{v(\boldsymbol{x})}$ (Corollary 1).  This means that when introducing $\boldsymbol{u(x)}$ during the quantization reconstruction, we can obtain a flat model on the direction of $\boldsymbol{v(x)}$. This inspires us to design a random dropping of activation quantization to cover as many cases of $\boldsymbol{u(x)}$ for flatness across different directions.
>
>   Besides rigorous proof in Appendix B, we also provide experimental evidence to support our analysis. The sharpness measurement in Fig. 2 and Hessian information in Appendix C.3 prove that (1) suitably introducing activation indeed helps optimize a flat quantized model. (2) QDrop indeed achieves a more general flatness by covering as many directions as possible randomly and thus fulfills the best performance.

---

### Official Review · Reviewer_iK8W · 2021-11-04

**Correctness:** 4
**Technical Novelty And Significance:** 3
**Empirical Novelty And Significance:** 3
**Recommendation:** 8
**Confidence:** 4

**Main Review:**

The paper presents a strikingly simple approach to activation quantization. It boils down to a per layer modified dropout of the quantized activation, where either the network backdrops through the quantized activation or the full precision activation. Since approach is developed and implemented via layer wise fine-tuning, we could say that the approach is "greedy" but with the benefit of being palpably easy to implement in practice.

Although the empirical results provided by the authors show that in practice the approach works well enough, I would like to see the authors to push the envelope further and try to use 1st or 2nd order gradient information (i.e. gradient or hessian) to better inform when applying their dropout is actually necessary (and to be even more clear, I'm thinking things like the Hessian criteria presented in HAWQ-v2 which the authors reference in their argumentation in favor of flatter models in Section 2 or the Gradient loss presented by Lee et al. in "Data-free mixed-precision quantization using novel sensitivity metric" https://arxiv.org/abs/2103.10051).

I would also like to author to explain how their approach relates and differs to the one introduced by Fan et al. at ICLR21 in "TRAINING WITH QUANTIZATION NOISE FOR EXTREME MODEL COMPRESSION" since both approaches seemed to boil down fine-tuning the model under a noise quantization modeling scheme. I don't necessarily need a numerical comparison, but simply a brief mention in the related work section would be sufficient to better inform the reader of the potential connection between these 2 works.

Overall, I'm satisfied with the paper, and I think it's in a good enough shape to be published at ICLR.


**Summary Of The Paper:**

The authors conduct theoretical studies on how activation quantization affects weight tuning, and their conclusion is that involving activation quantization into the reconstruction helps the flatness of model on calibration data and dropping partial quantization contributes to the flatness on test data. They present both empirical and theoretical findings, and propose the QDROP algorithm to exploit this phenomenon. QDROP randomly drops quantization during post-training quantization (PTQ) reconstruction to pursue the flatness from a general perspective. The authors also claim a new state of the art for PTQ on various tasks including image classification, object detection for computer vision, and text classification and question answering for natural language processing.

**Summary Of The Review:**

The paper presents a simple approach to post training activation quantization that should be easy to implement and test by other researchers. The authors present both a mathematical treatment and empirical results to back up their claims and I wasn't able to find any glaring error. Hence, I think the paper is good and should be accepted. I expect their code to be jointly published with their paper.

---

> ### Author Response · Authors · 2021-11-18
> **Response to Reviewer iK8W**
>
> Thank you for your insights and positive feedback on this paper. Below is the detailed response to each question, hope you can find them helpful.
>
> - Q1: "I would like to see the authors to push the envelope further and try to use 1st or 2nd order gradient information (i.e. gradient or hessian) to better inform when applying their dropout is actually necessary (and to be even more clear, I'm thinking things like the Hessian criteria presented in HAWQ-v2 which the authors reference in their argumentation in favor of flatter models in Section 2 or the Gradient loss presented by Lee et al. in "Data-free mixed-precision quantization using novel sensitivity metric" https://arxiv.org/abs/2103.10051)."
>
>   A: Thanks for your valuable suggestion. Leveraging the PyHessian [1], we calculate the top-1, top-5 highest Hessian eigenvalues ($\lambda_1$, $\lambda_5$), and the Hessian trace ($Tr$) under ResNet-18 W3A3 setting to compare the flatness of quantized models among the three Cases in Fig. 1 and QDrop. The detailed results are listed in the following table. We can see that QDrop has the lowest Hessian eigenvalue and trace among all the cases, which indicates that our QDrop has a flatter loss landscape. Together with our results (Fig. 3) in the paper, we provide both qualitative and quantitative evidence of the flatness in QDrop. We have added this Hessian comparison into our paper and will cover more settings in the future.
>
>   | Method | $\lambda_1$ | $\lambda_5$ | $Tr$   |
>   | ------ | ----------- | ----------- | ------ |
>   | Case 1 | 14770       | 6746        | 122894 |
>   | Case 2 | 8423        | 4050        | 86287  |
>   | Case 3 | 8258        | 3821        | 84044  |
>   | QDrop  | 6850        | 3044        | 66371  |
>
> - Q2: "I would also like to author to explain how their approach relates and differs to the one introduced by Fan et al. at ICLR21 in "TRAINING WITH QUANTIZATION NOISE FOR EXTREME MODEL COMPRESSION" since both approaches seemed to boil down fine-tuning the model under a noise quantization modeling scheme. I don't necessarily need a numerical comparison, but simply a brief mention in the related work section would be sufficient to better inform the reader of the potential connection between these 2 works."
>
>   A: Thanks for pointing out the relevant paper [2]. We find that the two papers have different motivations and focus on different topics. QDrop is motivated by the flatness property of the quantized model in PTQ and [2] is motivated by induced bias brought by STE in QAT.  Both papers introduce randomness to improve performance. But QDrop randomly drops activation quantization to pursue flat quantized models and [2] randomly adds noise to weight to simulate weight quantization. Therefore the two random noise schemes work in different ways. In our revision, we cite this paper and discuss the relationship and differences.
>
> - Q3: "I expect their code to be jointly published with their paper."
>
>   A: Thanks for your suggestion. We provided a version of our code in the supplementary materials during the first submission. We have open-sourced it via Github ([link](https://github.com/iclr22submission/QDrop)) as suggested. Hope that can be beneficial for the PTQ community.
>
> [1] Yao Z, Gholami A, Keutzer K, et al. Pyhessian: Neural networks through the lens of the hessian[C]//2020 IEEE International Conference on Big Data (Big Data). IEEE, 2020: 581-590.
>
> [2] Fan A, Stock P, Graham B, et al. Training with quantization noise for extreme model compression[J]. arXiv preprint arXiv:2004.07320, 2020.

---

### Author Response · Authors · 2021-11-18
**General Response**

We would like to thank all the reviewers for their constructive feedback. We are delighted to see that reviewers acknowledge our pioneer observations on the effect of activation quantization in PTQ tuning, the significant finding that flatness is critical for PTQ performance, and the SOTA results brought by our simple yet effective method across various tasks.

We will reply to your comments one by one, but first of all, we'd like to have a general response, clarifying some common questions:

- Regarding Fig. 3, we are really sorry for missing the detailed explanation. In this figure, we plot the loss landscape of the different Cases and our method, where we show QDrop reaches a flatter minimum.

- Some of the reviewers suggest a clearer illustration of our theoretical analysis and our method for better understanding. For further clarification, we reorganize the proof framework with necessary lemmas in the Appendix to guide the readers following it step by step. To summarize, we prove that:

  - the noise $\boldsymbol{u(x)}$ brought by activation quantization can be transformed into weight noise $\boldsymbol{v(x)}$ in the quantization background for both the fully connected and convolutional networks. —— Lemma 1
  - Thus with the activation quantization, we optimize an extra flatness term of the quantized model on perturbation $\boldsymbol{v(x)}$, which is helpful for pursuing optimal weights on calibration data. —— Theorem 1 & Corollary 1

  Based on the above theoretical findings, we conduct further analyses:

  Since the perturbation $\boldsymbol{v(x)}$ relates to input data, there is a flatness gap between calibration and test for PTQ, especially with limited calibration samples. To further improve the flatness on test data, we propose the method to cover more directions of perturbation and thus can diversify the flatness directions with little calibration data. Finally, the flatness on test data achieves an enhancement, which contributes to a significant accuracy boost. —— QDrop

We have updated our manuscripts, the change we made includes:

1. We reorganize the proofs in Appendix  B in revision to make the paper clearer.
2. We add discussions about Fig. 3 in Sec. 3.3 and also leverage Hessian information to support our analysis in Appendix C.3.
3. We add more discussion of actual meaning about our multiplicative form noise in Appendix A.
4. We take more technical terms to clarify the relationship between Eq. (8) and Corollary 1 more clearly.

For the detailed explanation, please see the response to each reviewer.

---

### Decision · Program_Chairs · 2022-01-20

**Decision:**

Accept (Poster)

**Comment:**

This paper proposes a simple, theoretically motivated approach for post-training quantization. The authors justify its effectiveness with both a sound theoretical analysis, and strong empirical results across many tasks and models, including a state-of-the-art result for 2-bit quantized weights/activations. All reviewers agreed the paper is worth accepting, with 3/4 rating it as a clear accept following the discussion period, and the fourth reviewer not giving strong reasons not to accept.